



# Killing the predator: impacts of top-predator mortality on global-ocean ecosystem structure

David Talmy[1] , Eric Carr[1], Harshana Rajakaruna[2], Selina Vage[3], Anne Willem-Omta[4]

[1]University of Tennessee Knoxville, Department of Microbiology, Knoxville, TN

[2]The Abigail Wexner Research Institute, Nationwide Children's Hospital, Columbus, OH

[3]The University of Bergen, Department of Biological Sciences, Norway

[4]Case Western Reserve University, Earth, Environmental, and Planetary Sciences

[*]Corresponding author: dtalmy@utk.edu

*Correspondence to*: David Talmy (dtalmy@utk.edu)

**Abstract.** Recent metanalyses suggest that microzooplankton biomass density scales linearly with phytoplankton biomass density, suggesting a simple, general rule may underpin trophic structure in the global ocean. Here, we use a set of highly simplified food-web models, solved within a global general circulation model, to examine the core drivers of linear predator-prey scaling. We examine a parallel food-chain model which assumes microzooplankton grazers feed on distinct size-classes of phytoplankton, and contrast this with a diamond food-

web model allowing shared microzooplankton predation on a range of phytoplankton size classes. Within these two contrasting model structures, we also evaluate the impact of fixed vs. density-dependent microzooplankton mortality. We find that the observed relationship between microzooplankton predators and prey can be reproduced with density-dependent mortality on the top predator, regardless of choices made about plankton food-web structure. Our findings point to the importance of parameterizing mortality of the top predator for models to

recapitulate trophic structure in the global ocean.

## 1.  Introduction

Over the past decades, there has been considerable progress in our understanding of marine planktonic ecosystems.

Both satellite and *in situ* observations have helped to elucidate the biogeography, phenology, and structure of these systems. Much of this knowledge has been incorporated in numerical models to make projections and perform sensitivity analyses, in particular pertaining to the impacts of global change (Dutkiewicz et al. 2013; Henson et al. 2021). As a result, marine ecosystem models have become increasingly detailed and complex, with a particular focus on improving the representation of the rich diversity of plankton. For example, the European Regional Seas





Ecosystem Model (ERSEM) contains 10 different plankton functional types and 3 types of bacteria (Butenschön et al. 2016), whereas the current version of the Plankton Type Ocean Model (PlankTOM11) includes 9 plankton functional types, bacteria, and jellyfish (Wright et al. 2021). The Darwin model uses allometric scaling to model dozens of plankton size classes (Ward et al. 2012; Henson et al. 2021)

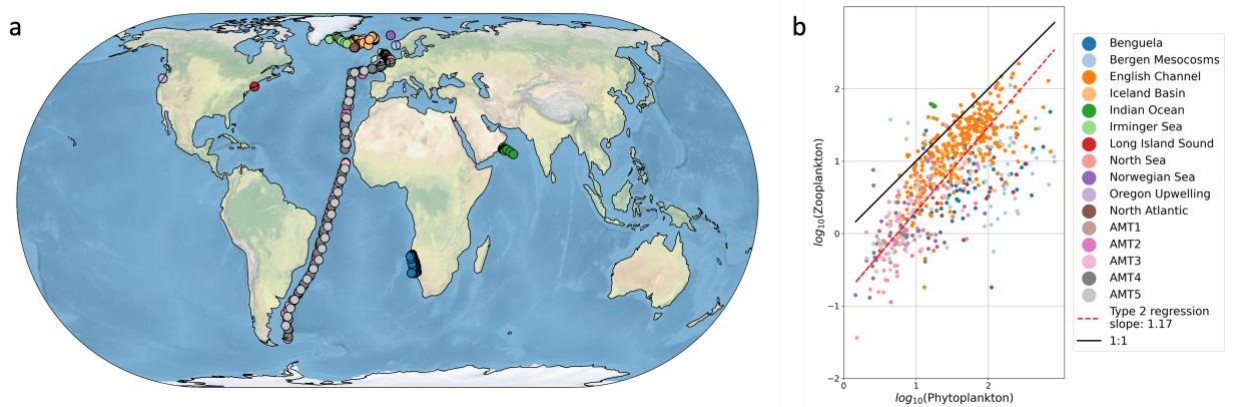

Figure 1. Recent metanalyses suggest microzoopankton biomass density scales linearly with phytoplankton biomass density in the global ocean a) sampling locations and b) relationship between microzooplankton and phytoplankton biomass in mg C m⁻³ (Rajakaruna et al. 2022).

As ecosystem models become increasingly complex, it becomes increasingly challenging to understand how their structure impacts the bulk biogeochemical properties of the system. For example, assumptions about microzooplankton predation on phytoplankton determines model predictions of phytoplankton carbon in the surface ocean, which in turn influences rates of carbon fixation, and eventually, carbon sequestration from the surface layer to the deep ocean. Due to their influence on carbon cycling globally, Earth system models typically

contain representations of ocean ecosystems, and are incorporating expanded plankton diversity (Séférian et al. 2020), raising questions about how much model complexity is required to capture biogeochemically relevant properties (Kwiatkowski et al. 2014).

Observational datasets provide a critical resource to discriminate between models with different assumptions about modeled food web interactions. The relationship between microbial predators and prey (for

example, microzooplankton and phytoplankton, respectively) is one observed phenomenon with profound implications for global biogeochemical cycles, for example by controlling the biomass of autotrophs that fix carbon,



and impacting carbon export through microzooplankton excretion of fecal pellets (Buck and Newton 1995). One recent metanalysis suggests a relatively simple set of observational relationships between microbial predators and their prey (Rajakaruna et al. 2022). Specifically, predator biomass (say, Y) appears to scale with prey biomass, (say, X), following a simple, linear relationship, i.e. Y~X (Figure 1). These observational compilations present the opportunity to identify key features of ocean biogeochemical models that capture relationships between predator and prey biomass.

Here, we undertake this task with a highly idealized set of ecosystem models, solved in the global ocean with a general circulation model. The models we examine are highly abstracted (Figure 2), capturing some essential features that are general to a wide class of ecosystem and biogeochemical models.

All ecosystem models with descriptions of diversity beyond the classic nutrient-phytoplankton-zooplankton-detritus (NPZD) formulation (Wroblewski 1989; Fasham et al. 1990), must make assumptions about which predators feed on which prey. However, it is unclear whether empirically rooted contrasting assumptions (Holt et al. 1995; Armstrong 1999) about predator preference for prey type impact the scaling between predator and prey biomass, in a manner that is consistent with patterns observed by Rajakaruna et al. (2022). Our models are put forth specifically to address this question.

In addition to asking whether food-web structure impacts plankton predator-prey relationships, we also consider the role of predation on the top predator, in this case the zooplankton. By their nature, planktonic ecosystem models do not explicitly resolve the dynamics of higher trophic levels. Therefore, the effects of higher predation on the top predator are usually parameterized (Steele and Henderson 1992; Edwards and Brindley 1999; Rhodes and Martin 2010). The choices made here often are not examined carefully with respect to the biogeochemical and ecological properties of the system.

By explicitly examining the role of predation on the top predator in the context of two contrasting food webs, we seek to identify the core, underlying drivers of linear scaling between microbial predators and prey (Figure 1). We then "sample" the model and compare predictions to observations of trophic structure covering a large range of temperate, subtropical, and tropical ecosystems. In doing so, we evaluate how these contrasting model structures impose trophic structure globally. While our ecosystem models are relatively simple by comparison to many extant biogeochemical models (e.g. Dutkiewicz et al. 2020), they are comparable to the ocean biology component of many extant Earth system models (Rohr et al. 2023), and allow clear insight. We discuss the implications of our findings for more complex models of ecosystem dynamics.





## 2. Models and Methods

In the sections that follow, we explain and justify the full equations used to parameterize phytoplankton and microzooplankton growth dynamics. We then describe model implementation in a global ocean ecosystem context and the comparison of simulations with a published compilation of relevant ocean data.

### 2.1 Food-web models


The main ecosystem model parameterization is reported in Table 1, and a schematic representation is provided in Figure 2. Model equations are described in detail in the Appendix, and our source code is provided in a publicly available repository (https://github.com/werdna-spatial/GUD_closure). We compared predictions of a model with 'diamond food-web' structure (shared predation), to a model assuming predators feed in parallel on distinct prey

types:

*Parallel food-chain model*. Here it was assumed that microzooplankton grazers feed in parallel on microzooplankton prey (Armstrong 1999; Ward et al. 2012, 2013), mimicking predation that is specific to different size classes or functional groups. Models with parallel feeding have led to realistic predictions of plankton

community composition in the global ocean (Ward et al. 2013; Dutkiewicz et al. 2020). Furthermore, parallel feeding is assumed in the biological component of many Earth system models (Rohr et al. 2023), making it a useful food-web structure to examine in the global ocean.

*Diamond food-web model*. An alternative to parallel feeding is a model with shared predation.  Here,

microzooplankton predators may feed on multiple plankton types. Since this general predation resembles a diamond, models with shared predation are referred to as having 'diamond' food-web structure (Holt et al. 1995). A recent study examining plankton community composition along a resource availability gradient in the North Pacific indicated a model with shared predation on *Prochlorococcus* and heterotrophic bacteria may in some circumstances lead to improved predictions of plankton community composition (Follett et al. 2022). Furthermore,

shared predation is assumed in the biological component of many Earth system models (Rohr et al. 2023), and it is therefore useful to examine the predictions of models that assume individual grazers prey on individual size classes of phytoplankton.

The parallel food chain and diamond food web models use established allometric scaling laws to assign traits
according to phytoplankton cell size (Banse 1976; Litchman et al. 2007; Ward et al. 2012; and see Appendix). In
both formulations, small and large phytoplankton represent cells with be ~0.5 and 7µm equivalent spherical radius,
and are representative of picocyanobacteria and eukaryotic algae, respectively. Small and large microzooplankton
represent protists ~3.2 and 4.7µm equivalent spherical radius and are representative of microzooplankton in the
ciliate size range. The generalist predator in the diamond food-web model has 3.2µm equivalent cell radius,
matching the small predator of the parallel model.

## 2.2 Parameterizing microzooplankton mortality

All lower trophic ecosystem models must make choices about the mortality of the top predator. Here, loss processes
must be mimicked, without being explicitly resolved. This requirement to parameterize presents a problem for
plankton ecosystem modelers wishing to motivate model form and function with mechanism. Nevertheless, one
way to evaluate the strength of different assumptions about model closure is to examine the influence of contrasting
assumptions on the model predictions in a holistic manner. Here, we sought to do this by applying two widely
assumed microzooplankton loss processes:

*Linear microzooplankton losses*. Here, it is assumed that the rate of microzooplankton mortality is independent of
its biomass density (Table 1). As such, linear losses may equivalently be thought of as density independent
mortality. This assumption has been applied within ecological and biogeochemical models to predict biogeography
of cyanobacteria and heterotrophic bacteria in the North Pacific (Follett et al. 2022), and in the global ocean (Ward
et al. 2012, 2013).

*Quadratic microzooplankton losses*. Here, the rate of microzooplankton mortality increases with biomass density.
This increase in mortality rate can be justified on several grounds, including intraspecific competition (Barbier and
Loreau 2019) and sinking (Schartau et al. 2007), which may both increase with microzooplankton density. Here,
we invoke density-dependent mortality on the highest trophic level to mimic the effects of unresolved predation on
the highest predator.





**Table 1.** Plankton ecosystem model equations.

Let the biomass of any plankton size class (either phytoplankton or microzooplankton) be represented generally $B_i$. Each of these biomass groups is constrained with mass balance equations for advection, mixing, sinking, and biological source and sink terms, as follows:

$$\frac{\partial B_i}{\partial t} = \underbrace{S_{B_i}}_{\substack{Biological \\ reactions}} - \underbrace{\nabla \cdot (\boldsymbol{u}B_i)}_{advection} + \underbrace{\nabla \cdot (\boldsymbol{\kappa}\nabla B_i)}_{diffusion} + \underbrace{\frac{\partial w_i B_i}{\partial z}}_{sinking}$$

Where $\boldsymbol{u}$ and $\boldsymbol{\kappa}$ velocity and diffusion coefficients, respectively, $w_i$ is a sinking speed, and $S_{B_i}$ represents all biological sources and sinks. Let $P_i$ and $Z_i$ represent any phytoplankton and microzooplankton size class, respectively. The planktonic sources and sinks for the diamond food-web and parallel model, respectively, are as follows:

Diamond food-web

$$S_{P_i} = \mu_i P_i - g_i Z_i - \delta_P P_i$$

(1)

$$S_{Z_1} = \varepsilon \left( \sum_{i=1,2} g_i \right) Z_1 - \delta_z Z_1 - \delta_{zz} Z_1^2$$

Parallel feeding

$$S_{P_i} = \mu_i P_i - g_i Z_i - \delta_P P_i$$

(2)

$$S_{Z_i} = \varepsilon g_i Z_i - \delta_z Z_i - \delta_{zz} Z_i^2$$



Where $\mu_i$ is the growth rate for each phytoplankton size class, that is sensitive to nutrient availability, light, and temperature. The functional sensitivity of $\mu_i$ to these environmental variables, along with mass balances for nutrients and detritus, are described in the Appendix. The grazing rate for each phytoplankton size class is $g_i$, and the proportion of material ingested into zooplankton biomass is $\varepsilon$. The mortality coefficients for linear and quadratic microzooplankton losses are $\delta_z$ and $\delta_{zz}$, respectively.

## 2.3 Global ocean ecosystem models

To explore the ecological and biogeochemical implications of these characteristics, we introduced these
parameterizations of primary and secondary producers into a global ocean ecosystem, biogeochemistry, and circulation model (MITgcm). The ecosystem model simulates flow of C, N, and other elements (Figure 2) between inorganic nutrients, photo-autotrophs, microzooplankton, and detritus. It is embedded in a coarse-resolution ($1° \times 1°$ horizontal, 24 vertical levels), climatologically averaged, global ocean circulation model that has been constrained with satellite and in situ observations (Wunsch and Heimbach 2007).


## 2.4 Model-data comparison and statistical analyses

We 'sampled' the model in locations where there are environmental samples in the compilation of Rajakaruna et al. (2022) (Figure 1). After log-transforming phytoplankton and microzooplankton biomass density, we conducted
ordinary least squares type 2 (OLS II) regression and quantified a Pearson correlation coefficient. We compared the regression slope and Pearson R value, between the models and the environmental datasets. To identify whether sampling locations were representative of the broader global ecosystem, we repeated the same analysis, sampling the entire global ocean. In doing so, we asked which model assumptions were necessary for the ecosystem model to reproduce internally the observed relationships between microzooplankton and phytoplankton biomass density
(Figure 1).



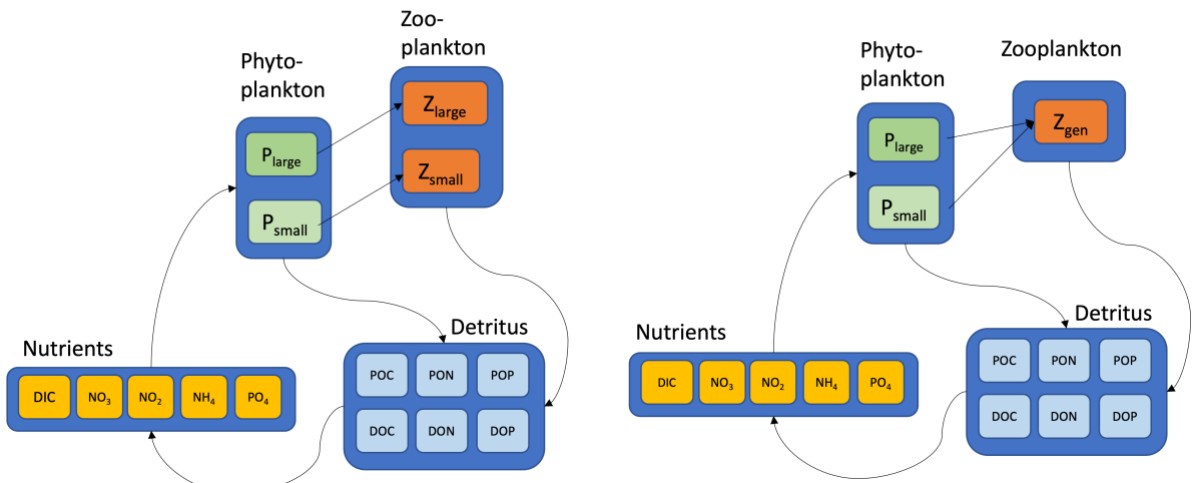

Figure 2: Two contrasting models considered here a) parallel food-chains b) shared predation in the form of a diamond food-web. Both models cycle elements (C, N and P) through inorganic and organic forms. Iron biogeochemistry was included in the model in a manner similar to C, N, and P but is not shown for parsimony. Model equations along with parameter definitions and units are detailed in the Appendix.

## 3. Results

We first describe model predictions of ecosystem structure in the global ocean and go on to examine which of the models leads to predictions of predator and prey biomass density consistent with the observations in Figure 1. In all that follows, we compare the predictions for both the diamond and parallel food chain models with and without density-dependent microzooplankton losses.

*3.1 Surface ocean phytoplankton carbon*. All models make qualitatively similar predictions of surface ocean total carbon (Figure 3). Phytoplankton carbon density is lowest in the low-latitude oligotrophic gyres, and highest in coastal regions, the equatorial upwelling, and at high latitude. These predictions are all qualitatively consistent with predictions of phytoplankton biomass density indicated by satellite remote sensing of ocean color (Hu et al. 2019). Interestingly, however, there are clear differences in the total phytoplankton carbon density predicted by the four
models, with the most notable contrast between the models with parallel feeding and the diamond food-web



(compare columns, Figure 3). Specifically, the model with parallel feeding tends to predict greater total phytoplankton carbon density at high latitude and in equatorial upwelling and coastal regions. There are more subtle increases in total phytoplankton carbon density when quadratic microzooplankton carbon density is assumed instead of linear microzooplankton mortality (compare bottom and top rows in Figure 3). These results identify a subtle interplay between food-web structure and microzooplankton mortality on predictions of phytoplankton carbon density in the global ocean.

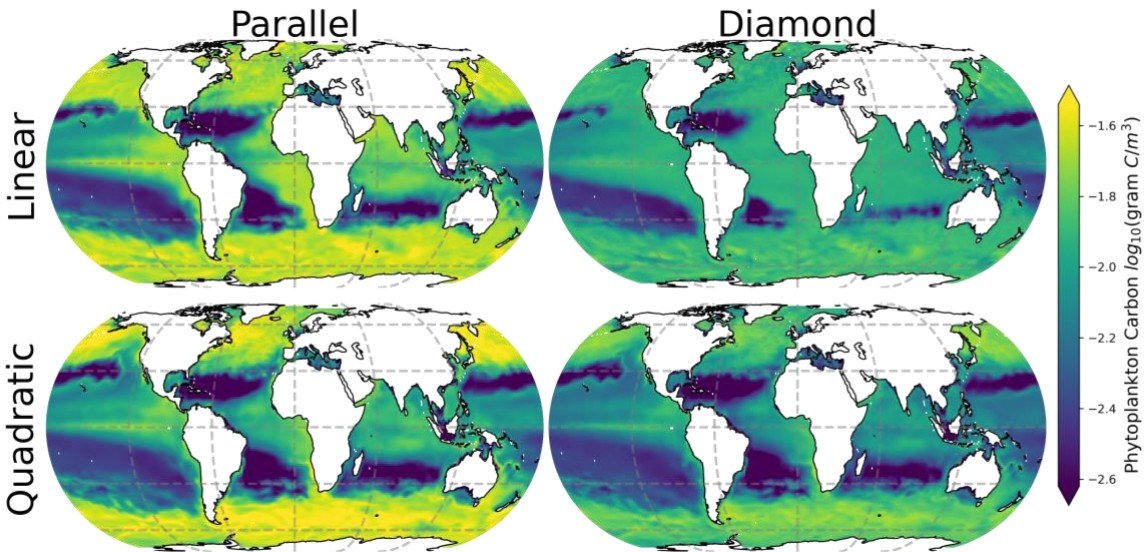

Figure 3. Total phytoplankton carbon in the surface ocean predicted by all four contrasting models. Color represents total phytoplankton carbon density for the surface ocean averaged over a seasonal cycle.

*3.2 Surface ocean community composition*. All four models predict qualitatively similar patterns in phytoplankton community composition in the surface ocean (Figure 4). Specifically, the small phytoplankton size-class dominates in the low-latitude oligotrophic gyres (deep red colors, Figure 4) and the large phytoplankton size class dominates at high latitudes (deep blue colors, Figure 4). Nevertheless, the model with shared predation (diamond food-web) predicts far greater competitive exclusion of the small phytoplankton size-class at high latitude. The parallel food-web model predicts coexistence of the small and large phytoplankton throughout much of the surface ocean, regardless of which microzooplankton closure is used (left-hand column, figure 4).





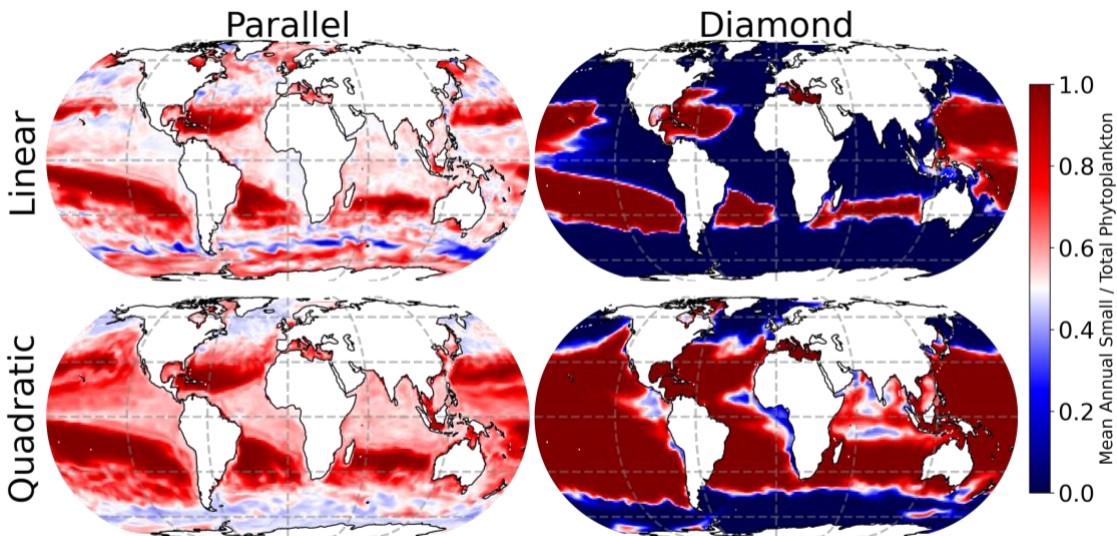

Figure 4. Phytoplankton community composition in the surface ocean. Red indicates dominance
of small phytoplankton; blue indicates dominance of large phytoplankton.

*3.3 Interplay between community composition and total carbon density.* Interestingly, the impact of food-web
structure (diamond vs. parallel) and microzooplankton closure (linear vs. quadratic) appear to be mirrored in the
model predictions of phytoplankton carbon density (Figure 3) and community composition (Figure 4). Specifically,
the greatest differences are between the diamond and parallel models (comparing columns) with more nuanced
differences between closure assumption (comparing rows). This mirroring points to community composition as a
driver of total phytoplankton carbon density. Specifically, anywhere in the ocean with greater representation of the
smaller size class tends to predict elevated total phytoplankton carbon density.

*3.4 Quadratic microzooplankton closure predicts global trophic structure.* The results in Figures 3-4 point to the
importance of food-web structure for predictions of planktonic ecosystem carbon in the global ocean. We now turn
our attention to ask, which of these models is consistent with observations that microzooplankton carbon density
scales linearly with phytoplankton carbon density (Figure 1, Rajakaruna et al., 2022).

In Figure 5, we show the relationship between total microzooplankton and phytoplankton carbon for the global
ocean. Each colored point represents the number of 1° grid cells falling within the microzooplankton and
phytoplankton carbon density marked by its position on the axes. The dashed black line represents the OLS II

regression slope. Interestingly, we find here that any differences between the parallel and diamond food-web are minimal (compare columns, Figure 5), and the largest differences are between the linear and quadratic microzooplankton closure (compare rows, Figure 5). Therefore, in predicting ecosystem trophic structure, which we think of here as the relationship between microzooplankton and phytoplankton carbon density, the impacts of food-web on phytoplankton community composition that were revealed in Figures 3-4 cease to play an important role. Moreover, only the model with quadratic microzooplankton losses predicts a relationship between microzooplankton and phytoplankton carbon density that is consistent with the linear scaling in the observation dataset in Figure 1 (bottom row, Figure 5). The model with linear microzooplankton mortality predicts far less correlation between microzooplankton and phytoplankton carbon density (reflected in the lower r values) and a shallower slope relating total microzooplankton carbon with phytoplankton carbon.

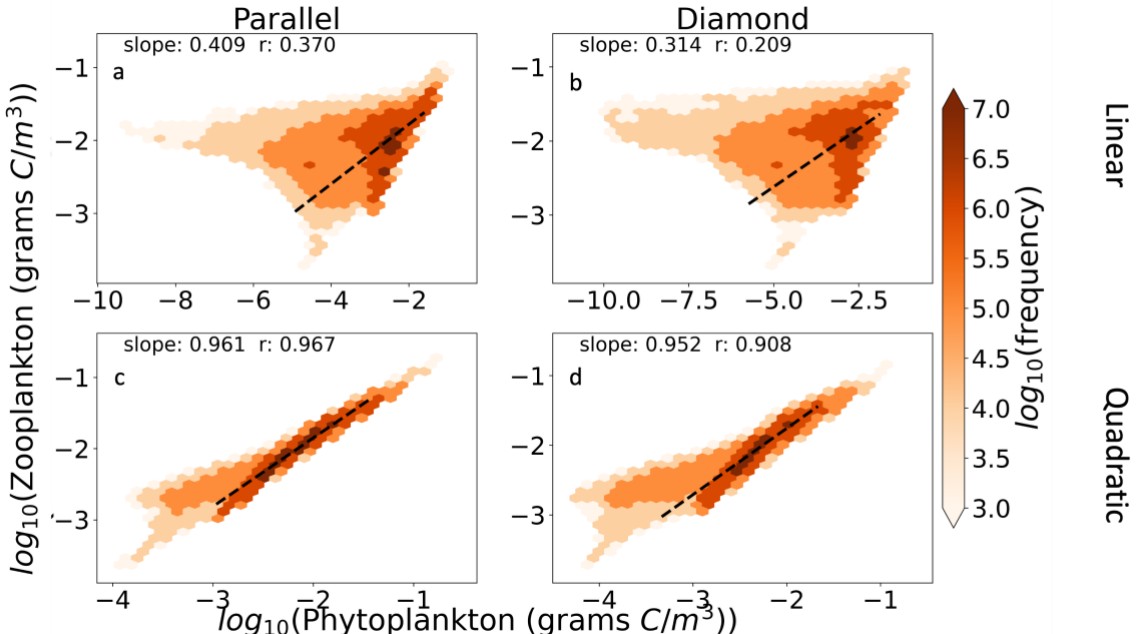

Figure 5: The relationship between phytoplankton and microzooplankton carbon density in the global ocean for a) parallel food chain model with linear closure b) diamond food-web with linear closure c) parallel food chain model with quadratic closure and d) diamond food-web model with quadratic closure. The color within each hexagon represents the number of 1° grid cells that fall within the biomass range marked by their position on the axes. Slopes are OLS Type 2 regression slopes and r values are Pearson correlation coefficients. Density-dependent microzooplankton mortality reproduces the relationship between Z and P, regardless of food-web structure.



Why does the linear closure predict such a variable relationship between microzooplankton and phytoplankton carbon density? To investigate these predictions, we attempted to separate out spatial and temporal impacts on the relationship.

In figure 6, we show seasonal variability in the relationship between microzooplankton and phytoplankton carbon density, for a single site in the English Channel. Here, the model with linear zooplankton losses predicts a cyclic relationship between phytoplankton and microzooplankton biomass, irrespective of parallel vs. diamond food-web structure. The regression slopes for this single location mirror the regression slopes for the global ocean – the linear microzooplankton closure predicts a shallow regression slope whereas the diamond food-web model predicts approximately linear scaling. Furthermore, the quadratic closure predicts far higher correlation between phytoplankton and microzooplankton biomass, again consistent with the global collection (Figure 5). These results point to the tendency of the linear closure to predict predator-prey oscillations as a key driver of the global relationship between phytoplankton and microzooplankton biomass density. The cyclic behavior is true irrespective of assumptions about parallel food-chains vs. a diamond food-web.



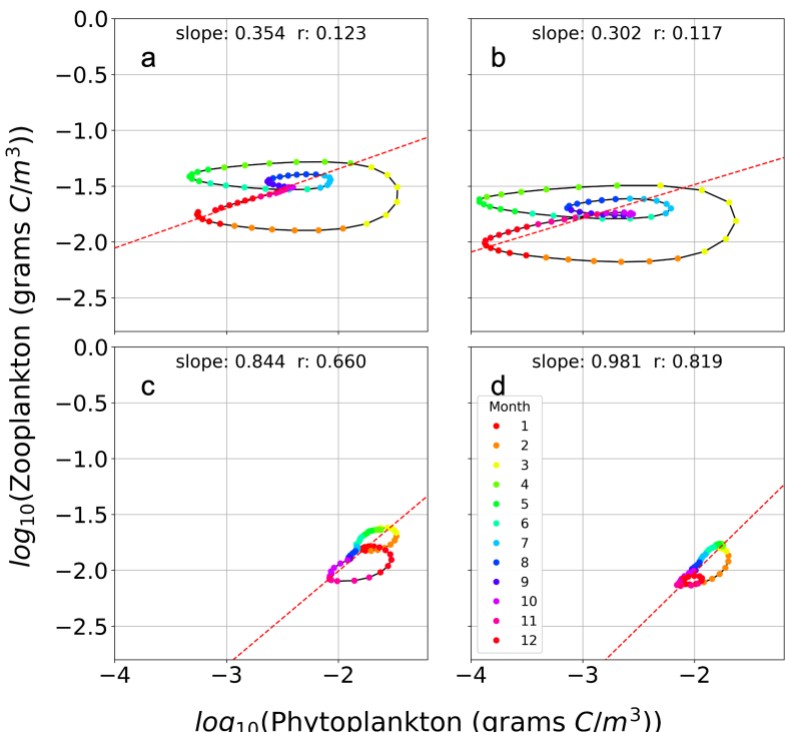

Figure 6: The relationship between total microzooplankton and phytoplankton carbon density during a seasonal cycle in the English Channel for a) parallel food chain model with linear closure b) diamond food-web with linear closure c) parallel food chain model with quadratic closure and d) diamond food-web model with quadratic closure. Linear losses on the microzooplankton predict cyclic behavior in the predator-prey relationship that are inconsistent with observations (Figure 1).

In Figure 7, we show spatial variation in the Z:P ratio in the surface ocean, where the color in each location represents the seasonally averaged Z:P ratio. Interestingly, there is considerable spatial variability in Z:P for either food-web assuming linear closure (top row, Figure 7), with the Z:P ratio rising at higher latitudes (top row, Figure 7). This prediction is consistent with prior estimates of Z:P variability in the global ocean that assumed linear closure and parallel feeding (Ward et al. 2012). The quadratic closure removes much of this spatial variation (note

the narrower color bar range in the bottom row, Figure 7). In steady-state, linear losses on the microzooplankton allows them to place a limit on the phytoplankton biomass, causing carbon to accumulate in the predator (Follett

et al. 2022). Density-dependent mortality on the microzooplankton forces the microzooplankton to be removed at a rate that is commensurate with their biomass density, inhibiting their ability to limit the phytoplankton population size, and causing both predators and prey to rise together as the system is enriched with resources.


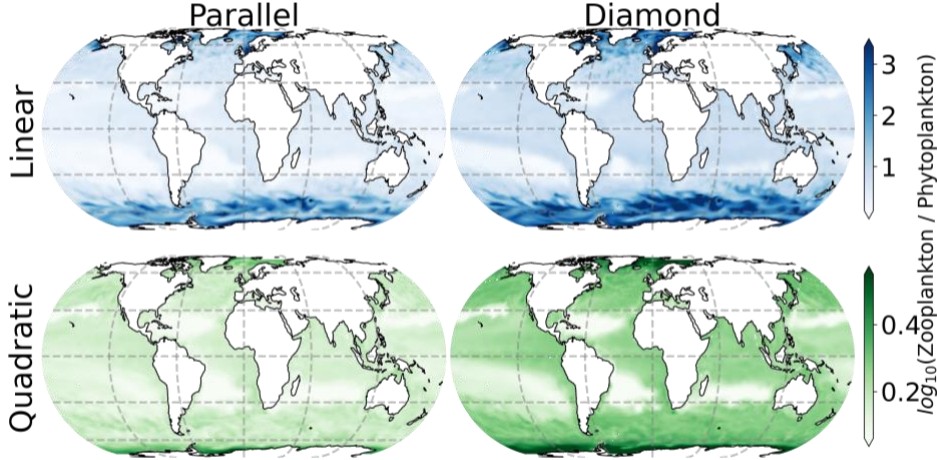

Figure 7. Seasonally averaged surface ocean Z:P ratio. Spatial variability in Z:P ratio is lessened by quadratic microzooplankton losses, irrespective of food-web structure.

## 4.  Discussion

Microzooplankton predation on phytoplankton determines phytoplankton carbon in the surface ocean, which in turn influences rates of carbon fixation, and eventually, carbon sequestration from the surface layer to the deep ocean.  Models of plankton ecosystem structure are becoming increasingly complex, but models with relatively simple representation of plankton food-webs are important components of many extant Earth system models (Rohr et al. 2023). Here, we have examined a set of models with minimal complexity, in the context of environmental data, to examine core drivers of system structure globally. We find that total phytoplankton carbon density, and community composition, are profoundly impacted by choices regarding food-web structure and losses on the highest predator (in this case, microzooplankton grazers). The diamond food-web predicts competitive exclusion of small and large phytoplankton size-classes, whereas parallel feeding allows the small phytoplankton size class to persist throughout much of the surface ocean, during high latitude blooms and in coastal upwelling regions.





Persistence of small phytoplankton size-classes at higher latitudes is consistent with observational data showing
pico (<2µm) and nano plankton (2-20 µm) persist through temperature and resource gradients in a wide range of
ocean environments (Marañón et al. 2012). These findings, in tandem with our study and prior modeling studies
(Ward et al. 2012, 2013), points to parallel feeding as a pervasive influence on planktonic system structure.
Nevertheless, shared predation has been invoked to explain *Prochlorococcus* die-off with latitude (Follett et al.
2022) and is invoked in many extant Earth system models (Rohr et al. 2023). Therefore, both food-web structures
considered here (parallel feeding and the diamond food-web) may exist in natural planktonic systems, and are also
assumed within models of the global ocean that inform climate change projections. Our findings point to the need
to carefully consider assumptions about predation on the highest trophic level with application of either model
structure, since these have profound implications both for phytoplankton carbon inventories, and community
composition.

By comparison to the models considered here, plankton communities are considerably more complex and
diverse, regarding organism size (Hansen et al. 1994), metabolism (Alexander et al. 2015; Posfai et al. 2017) and
resource affinities (Litchman et al. 2007). Furthermore, neither representation of microzooplankton grazing
considered here provides a realistic view of planktonic feeding, which can involve microzooplankton switching
between prey types, with implications for community composition and system structure (Vallina et al. 2014). Our
modeling could have been made considerably more complex to evaluate these more realistic influences on system
structure. Similarly, when it comes to parameterizing microzooplankton losses, there are more complex
assumptions to be made beyond our very crude contrast between linear (density-independent) and quadratic
(density-dependent) mortality, again with implications for system properties (Rhodes and Martin 2010; Omta et al.
2023).

Despite these limitations, our modeling points to a simple set of principles that we anticipate will extend
to more sophisticated representations of plankton ecology. In particular, the quadratic microzooplankton closure
provides a realistic and important constraint on the relationship between microzooplankton and phytoplankton
carbon density, irrespective of the assumed food-web. This generality and consistency with observational data may
also apply to other predator-prey interactions. Linear scaling between predators and prey abundance has also been
observed between viruses and heterotrophic bacteria (Rajakaruna et al. 2022). Viral infection is thought to be highly
host-specific (Flores et al. 2011) suggesting parallel food-web structure between predators and prey may be more
appropriate. Our finding that linear scaling between predators and prey can be reproduced with the quadratic
closure, regardless of food-web structure, provides insight that may inform models of plankton ecosystems that
include even more diverse representations of microbial life.






## **Appendix A: Model description.**

Here we provide details of the ecosystem model represented graphically in Figure 2. The description is very similar to other implementations of the Darwin ecosystem model (Dutkiewicz et al. 2009, 2012; Ward et al. 2012; Zakem

et al. 2018). All model parameter variable definitions and units are provided in Tables A1-A3.

Inorganic nutrients such as ammonium are governed by a mass balance for advection, diffusion, and biological sources and sinks:

$$\frac{\partial NH_4^+}{\partial t} = \underbrace{S_{NH_4^+}}_{\substack{Biological \\ reactions}} - \underbrace{\nabla \cdot (\boldsymbol{u} NO_3^-)}_{advection} + \underbrace{\nabla \cdot (\boldsymbol{\kappa} \nabla NO_3^-)}_{diffusion} \tag{A1}$$

Tracers describing nitrate, nitrite, phosphate, dissolved inorganic carbon, and iron are described similarly:

$$\frac{\partial NO_3^-}{\partial t} = S_{NO_3^-} - \nabla \cdot (\boldsymbol{u} NO_3^-) + \nabla \cdot (\boldsymbol{\kappa} \nabla NO_3^-) \tag{A2}$$

$$\frac{\partial NO_2^-}{\partial t} = S_{NO_2^-} - \nabla \cdot (\boldsymbol{u} NO_2^-) + \nabla \cdot (\boldsymbol{\kappa} \nabla NO_2^-) \tag{A3}$$

$$\frac{\partial PO_4^{3-}}{\partial t} = S_{PO_4^{3-}} - \nabla \cdot (\boldsymbol{u} PO_4^{3-}) + \nabla \cdot (\boldsymbol{\kappa} \nabla PO_4^{3-}) \tag{A4}$$

$$\frac{\partial DIC}{\partial t} = S_{DIC} - \nabla \cdot (\boldsymbol{u} DIC) + \nabla \cdot (\boldsymbol{\kappa} \nabla DIC) \tag{A5}$$

$$\frac{\partial Fe}{\partial t} = S_{Fe} - \nabla \cdot (\boldsymbol{u} Fe) + \nabla \cdot (\boldsymbol{\kappa} \nabla Fe) \tag{A6}$$

Pools of dissolved organic material for carbon, nitrogen, phosphorus, and iron are governed by a similar mass balance:

$$\frac{\partial DOC}{\partial t} = S_{DOC} - \nabla \cdot (\boldsymbol{u} DOC) + \nabla \cdot (\boldsymbol{\kappa} \nabla DOC) \tag{A7}$$

$$\frac{\partial DON}{\partial t} = S_{DON} - \nabla \cdot (\boldsymbol{u} DON) + \nabla \cdot (\boldsymbol{\kappa} \nabla DON) \tag{A8}$$





$$\frac{\partial DOP}{\partial t} = S_{DOP} - \nabla \cdot (\boldsymbol{u}DOP) + \nabla \cdot (\boldsymbol{\kappa}\nabla DOP) \tag{A9}$$

$$\frac{\partial DOFe}{\partial t} = S_{DOFe} - \nabla \cdot (\boldsymbol{u}DOFe) + \nabla \cdot (\boldsymbol{\kappa}\nabla DOFe) \tag{A10}$$

As are pools of particulate detritus for these three elements:

$$\frac{\partial POC}{\partial t} = S_{POC} - \nabla \cdot (\boldsymbol{u}POC) + \nabla \cdot (\boldsymbol{\kappa}\nabla POC) - \frac{\partial w_{POC}POC}{\partial z} \tag{A11}$$

$$\frac{\partial PON}{\partial t} = S_{PON} - \nabla \cdot (\boldsymbol{u}PON) + \nabla \cdot (\boldsymbol{\kappa}\nabla PON) - \frac{\partial w_{PON}POC}{\partial z} \tag{A12}$$

$$\frac{\partial POP}{\partial t} = S_{POP} - \nabla \cdot (\boldsymbol{u}POP) + \nabla \cdot (\boldsymbol{\kappa}\nabla POP) - \frac{\partial w_{POP}POC}{\partial z} \tag{A13}$$

$$\frac{\partial POFe}{\partial t} = S_{POFe} - \nabla \cdot (\boldsymbol{u}POFe) + \nabla \cdot (\boldsymbol{\kappa}\nabla POFe) - \frac{\partial w_{POFe}POFe}{\partial z} \tag{A14}$$

where particles are assumed to also sink at rate $w_{POC}, w_{PON}, w_{POP}, w_{POFe}$ for carbon, nitrogen, phosphorus, and iron, respectively.

Several of the biological source and sink terms are described in the main text (Table 1). Here we describe additional source and sink terms for inorganic nutrients and detritus. Ammonium is produced by remineralization of organic material, and lost by nitrification and phytoplankton growth:

$$S_{NH_4^+} = \underbrace{r_{DON}DON + r_{PON}PON}_{remineralization} - \underbrace{\zeta_{NH_4^+}NH_4^+}_{nitrification} - \underbrace{\sum_j V_{NH_4^+}P_j}_{nutrient\ uptake} \tag{A15}$$

Biological source and sink terms for nitrate, nitrite, phosphorus, and dissolved inorganic carbon are as follows:

$$S_{NO_3^-} = \underbrace{\zeta_{NO_2^-}NO_2^-}_{nitrification} - \underbrace{Q_{N:C}\sum_j V_{NO_3^-,j}P_j}_{nitrate\ uptake} \tag{A16}$$

$$S_{NO_2^-} = \underbrace{\zeta_{NH_4^+}NH_4^+}_{nitrification} - \underbrace{\zeta_{NO_2^-}NO_2^-}_{nitrification} - \underbrace{Q_{N:C}\sum_j V_{NO_2^-,j}P_j}_{nitrite\ uptake} \tag{A17}$$

$$S_{PO_4^{3-}} = \underbrace{r_{DOP}DOP + r_{POP}POP}_{remineralization} - \underbrace{Q_{P:C}\sum_j V_{DIC,j}P_j}_{phosphate\ uptake} \tag{A18}$$



$$S_{DIC} = \underbrace{r_{DOC}DOC + r_{POC}POC}_{remineralization} - \underbrace{\sum_j V_{DIC,j}P_j}_{DIC\ uptake} \tag{A19}$$

$$S_{Fe} = \underbrace{r_{DOFe}DOFe + r_{POFe}POFe}_{remineralization} - \underbrace{Q_{Fe:C}\sum_j V_{DIC,j}P_j}_{iron\ uptake} \tag{A20}$$

Where fixed elemental ratios convert carbon uptake to other elements (e.g. multiplication by $Q_{P:C}$ in Equation A18 converts carbon uptake to phosphorus uptake).

Dissolved organic material, for example DOC, is produced through phytoplankton and zooplankton mortality and sloppy feeding, and consumed through remineralization:

$$S_{DOC} = \underbrace{\sum_i (1-\beta_p^{mort})\,\delta_p P_i}_{phytoplankton\ mortality} + \underbrace{\sum_j (1-\beta_z^{mort})(\delta_z Z_j + \delta_{zz}Z_j^2)}_{zooplankton\ mortality} \tag{A21}$$

$$+ \underbrace{\sum_i \sum_j (1-\beta_z^{graz})(1-\varepsilon)g_i Z_j}_{sloppy\ feeding} - \underbrace{r_{DOC}DOC}_{remineralization}$$

Dissolved nitrogen and phosphorus are governed by the same sources and sinks, converted from carbon with fixed stoichiometric ratios, e.g. for nitrogen $Q_{N:C}$ (units mol N (mol C)$^{-1}$):

$$S_{DON} = Q_{N:C}\underbrace{\sum_i (1-\beta_p^{mort})\,\delta_p P_i}_{phytoplankton\ mortality} + Q_{N:C}\underbrace{\sum_j (1-\beta_z^{mort})(\delta_z Z_j + \delta_{zz}Z_j^2)}_{zooplankton\ mortality} \tag{A22}$$

$$+ Q_{N:C}\underbrace{\sum_i \sum_j (1-\beta_z^{graz})(1-\varepsilon)g_i Z_j}_{sloppy\ feeding} - \underbrace{r_{DON}DON}_{remineralization}$$

The same basic processes are also biological sources and sinks for particulate organic carbon:

$$S_{POC} = \underbrace{\sum_i \beta_p^{mort}\,\delta_p P_i}_{phytoplankton\ mortality} + \underbrace{\sum_j \beta_z^{mort}(\delta_z Z_j + \delta_{zz}Z_j^2)}_{zooplankton\ mortality} \tag{A23}$$

$$- \underbrace{\sum_i \sum_j \beta_z^{graz}(1-\varepsilon)g_i Z_j}_{sloppy\ feeding} - \underbrace{r_{POC}POC}_{remineralization}$$

Where $\beta_p^{mort}$ and $\beta_Z^{mort}$ partitions phytoplankton and zooplankton losses between particulate and dissolved pools, 320 with corresponding partitions for sloppy feeding given by $\beta_p^{graz}$ and $\beta_Z^{graz}$. As with DOM (Equation A22), fixed





stoichiometric conversations are applied to convert carbon POC sources to PON and POP. These equations are not shown for brevity.

The phytoplankton growth rate $\mu_i$ is modified by light, nutrients, and temperature in a multiplicative manner:

$$\mu_i = \mu_{max,i}\gamma_{L,i}\gamma_{N,i}\gamma_{T,i} \tag{A24}$$


Where light limitation is based on the model of photoacclimation following Geider et al. (1997):

$$\gamma_{L,i} = \left(1 - \exp\left(\frac{-\alpha\theta I}{\mu_{max,i}\gamma_{N,i}\gamma_{T,i}}\right)\right) \tag{A25}$$

Nutrient limitation follows Monod kinetics and Liebig's law of the minimum:

$$\gamma_N = min\{V_{N,i}, V_{P,i}, V_{Fe,i}\} \tag{A26}$$

Where nutrient limitation by nitrogen, phosphorus, and iron are governed by monod kinetics:

$$V_{N,i} = \frac{NO_3^-}{NO_3^- + K_{NO_3^-}}e^{\Psi NH_4^+} + \frac{NO_2^-}{NO_2^- + K_{NO_2^-}}e^{\Psi NH_4^+} + \frac{NH_4^+}{NH_4^+ + K_{NH_4^+}} \tag{A27}$$

$$V_{P,i} = \frac{PO_4^{3-}}{PO_4^{3-} + K_{PO_4^{3-}}} \tag{A28}$$

$$V_{Fe,i} = \frac{Fe}{Fe + K_{Fe}} \tag{A29}$$

Where nitrate and nitrite assimilation are inhibited in the presence of ammonium with $\Psi$, following (Follows et al. 2007) and others l (Dutkiewicz et al. 2009, 2012; Ward et al. 2012; Zakem et al. 2018). Uptake of ammonium, nitrite, and nitrate are found by partitioning total realized nutrient uptake by the three different nitrogen species as

follows:

$$V_{NH_4^+,i} = \frac{1}{V_{N,i}}\frac{NH_4^+}{NH_4^+ + K_{NH_4^+}}\gamma_N \tag{A30}$$

$$V_{NO_2^-,i} = \frac{1}{V_{N,i}}\frac{NO_2^-}{NO_2^- + K_{NO_2^-}}e^{\Psi NH_4^+}\gamma_N \tag{A31}$$



$$V_{NO_3^-,i} = \frac{1}{V_{N,i}} \frac{NO_3^-}{NO_3^- + K_{NO_3^-}} e^{\Psi NH_4^+} \gamma_N \tag{A32}$$

Growth is modulated by temperature with the Arrhenius equation:

$$\gamma_T = \tau \exp\left( A_E \left( \frac{1}{T + 273.15} - \frac{1}{T_0} \right) \right) \tag{A33}$$

Grazing rate of zooplankton type $j$ follows the Holling II functional response as a function of total phytoplankton biomass (Holling 1959), partitioned between phytoplankton size classes according to the proportion of total phytoplankton biomass in each size class:

$$g_{i,j} = g_{max,j} \frac{P_i}{\sum_i P_i} \frac{\sum_i P_i}{\sum_i P_i + K_{g,j}} \tag{A34}$$




Table A1. Model state variables.

| Symbol | Description | Units |
|---|---|---|
| $NH_4^+$ | Ammonium | mmol m$^{-3}$ |
| $NO_3^-$ | Nitrate | mmol m$^{-3}$ |
| $NO_2^-$ | Nitrite | mmol m$^{-3}$ |
| $PO_4^{3-}$ | Phosphate | mmol m$^{-3}$ |
| $DIC$ | Dissolved inorganic carbon | mmol m$^{-3}$ |
| $Fe$ | Iron | mmol m$^{-3}$ |
| $DOC$ | Dissolved organic carbon | mmol m$^{-3}$ |
| $DON$ | Dissolved organic nitrogen | mmol m$^{-3}$ |
| $DOP$ | Dissolved organic phosphorus | mmol m$^{-3}$ |
| $DOFe$ | Dissolved organic iron | mmol m$^{-3}$ |
| $POC$ | Particulate organic carbon | mmol m$^{-3}$ |
| $PON$ | Particulate organic nitrogen | mmol m$^{-3}$ |
| $POP$ | Particulate organic phosphorus | mmol m$^{-3}$ |
| $POFe$ | Particulate organic iron | mmol m$^{-3}$ |





Table A2. Biological source and sink variables.


| Symbol | Description | Units |
|---|---|---|
| $S_{NH_4^+}$ | Biological sources and sinks of ammonium | mmol m$^{-3}$ s$^{-1}$ |
| $S_{NO_3^-}$ | Biological sources and sinks of nitrate | mmol m$^{-3}$ s$^{-1}$ |
| $S_{NO_2^-}$ | Biological sources and sinks of nitrite | mmol m$^{-3}$ s$^{-1}$ 355 |
| $S_{PO_4^{3-}}$ | Biological sources and sinks of phosphate | mmol m$^{-3}$ s$^{-1}$ |
| $S_{DIC}$ | Biological sources and sinks of DIC | mmol m$^{-3}$ s$^{-1}$ |
| $S_{Fe}$ | Biological sources and sinks of iron | mmol m$^{-3}$ s$^{-1}$ |
| $S_{DOC}$ | Biological sources and sinks of DOC | mmol m$^{-3}$ s$^{-1}$ 360 |
| $S_{DON}$ | Biological sources and sinks of DON | mmol m$^{-3}$ s$^{-1}$ |
| $S_{DOP}$ | Biological sources and sinks of DOP | mmol m$^{-3}$ s$^{-1}$ |
| $S_{DOFe}$ | Biological sources and sinks of DOFe | mmol m$^{-3}$ s$^{-1}$ |
| $S_{POC}$ | Biological sources and sinks of POC | mmol m$^{-3}$ s$^{-1}$ |
| $S_{PON}$ | Biological sources and sinks of PON | mmol m$^{-3}$ s$^{-1}$ 365 |
| $S_{POP}$ | Biological sources and sinks of POP | mmol m$^{-3}$ s$^{-1}$ |
| $S_{POFe}$ | Biological sources and sinks of POFe | mmol m$^{-3}$ s$^{-1}$ |







Table A3. Model parameters and variables. Specific parameter values assume default values listed in various publications (Ward et al. 2012; Dutkiewicz et al. 2020) and are available in the online documentation for the Darwin ecosystem model (https://darwin3.readthedocs.io/en/latest/phys_pkgs/darwin.html). Plankton traits (nutrient and grazing half-saturation constants, maximal grazing and nutrient uptake rates) were generated via allometric scaling relationships reported by Ward et al. (2012). The large phytoplankton has a faster maximal growth rate and higher nutrient half-saturation constants than the small phytoplankton, representative of differences in growth rate between a eukaryotic algae and a cyanobacteria, respectively (Litchman et al. 2007; Ward et al. 2012)

| Symbol | Description | Units |
|---|---|---|
| $w_{POC}$ | Particulate organic carbon sinking rate | m s$^{-1}$ |
| $w_{PON}$ | Particulate organic nitrogen sinking rate | m s$^{-1}$ |
| $w_{POP}$ | Particulate organic phosphorus sinking rate | m s$^{-1}$ |
| $w_{POFe}$ | Particulate organic iron sinking rate | m s$^{-1}$ |
| $r_{DOC}$ | Particulate organic carbon remineralization rate | s$^{-1}$ |
| $r_{DON}$ | Particulate organic nitrogen remineralization rate | s$^{-1}$ |
| $r_{DOP}$ | Particulate organic phosphorus remineralization rate | s$^{-1}$ |
| $r_{DOFe}$ | Particulate organic iron remineralization rate | s$^{-1}$ |
| $r_{POC}$ | Dissolved organic carbon remineralization rate | s$^{-1}$ |
| $r_{PON}$ | Dissolved organic nitrogen remineralization rate | s$^{-1}$ |
| $r_{POP}$ | Dissolved organic phosphorus remineralization rate | s$^{-1}$ |
| $r_{POFe}$ | Dissolved organic iron remineralization rate | s$^{-1}$ |
| $\zeta_{NH_4^+}$ | Rate of ammonium oxidation to nitrite | s$^{-1}$ |
| $\zeta_{NO_2^-}$ | Rate of nitrite oxidation to nitrate | s$^{-1}$ |
| $V_{NH_4^+,i}$ | Rate of ammonium uptake by phytoplankton $i$ | s$^{-1}$ |
| $V_{NO_3^-,i}$ | Rate of nitrate uptake by phytoplankton $i$ | s$^{-1}$ |
| $V_{NO_2^-,i}$ | Rate of nitrite uptake by phytoplankton $i$ | s$^{-1}$ |





| $V_{DIC,i}$ | Rate of DIC uptake by phytoplankton $i$ | $s^{-1}$ |
| $Q_{N:C}$ | Phytoplankton ratio of nitrogen to carbon | mol N (mol C)$^{-1}$ |
| $Q_{P:C}$ | Phytoplankton ratio of phosphorus to carbon | mol P (mol C)$^{-1}$ |
| $Q_{Fe:C}$ | Phytoplankton ratio of iron to carbon | mol Fe (mol C)$^{-1}$ |
| $\beta_p^{mort}$ | Proportion of phytoplankton mortality that goes to POC | n.d. |
| $\beta_z^{mort}$ | Proportion of zooplankton mortality that goes to POC | n.d. |
| $\beta_z^{graz}$ | Proportion of sloppy feeding that goes to POC | n.d. |
| $\delta_p$ | Phytoplankton linear rate of mortality | $s^{-1}$ |
| $\delta_z$ | Zooplankton linear rate of mortality | $s^{-1}$ |
| $\delta_{zz}$ | Zooplankton quadratic rate of mortality | m$^3$ (mol C)$^{-1}$ s$^{-1}$ |
| $\mu_i$ | Growth rate of phytoplankton $i$ | $s^{-1}$ |
| $\mu_{max,i}$ | Maximum growth rate of phytoplankton $i$ | $s^{-1}$ |
| $\gamma_{L,i}$ | Growth limitation by light | n.d. |
| $\gamma_{N,i}$ | Growth limitation by nutrients | n.d. |
| $\gamma_{T,i}$ | Growth modulation by temperature | n.d. |
| $\theta$ | Phytoplankton chlorophyll to carbon ratio | mg Chl (mmol C)$^{-1}$ |
| $\alpha$ | Phytoplankton light affinity | m$^2$ mmol C (μmol photons)$^{-1}$ (mg Chl)$^{-1}$ |
| $I$ | Photosynthetically available radiance | μmol photons m$^{-2}$ s$^{-1}$ |
| $K_{NO_3^-,i}$ | Phytoplankton $i$ half-saturation constant for nitrate | mmol m$^{-3}$ |
| $K_{NO_2^-,i}$ | Phytoplankton $i$ half-saturation constant for nitrite | mmol m$^{-3}$ |
| $K_{NH_4^+,i}$ | Phytoplankton $i$ half-saturation constant for ammonium | mmol m$^{-3}$ |
| $K_{PO_4^{3-},i}$ | Phytoplankton $i$ half-saturation constant for phosphate | mmol m$^{-3}$ |
| $K_{Fe}$ | Phytoplankton $i$ half-saturation constant for iron | mmol m$^{-3}$ |
| $\Psi$ | Ammonium inhibition of nitrate and nitrite assimilation | m$^3$ (mmol N)$^{-1}$ |
| $A_E$ | Temperature response function coefficient | ºC |
| $\tau$ | Temperature response function coefficient | n.d. |
| $T_0$ | Reference temperature for Arrhenius growth response | ºC |





| $g_{max,i}$ | Maximal grazing rate of zooplankton $i$ | s$^{-1}$ | |
| --- | --- | --- | --- |
| $\varepsilon$ | Proportion of grazed phytoplankton assimilated by zooplankton | n.d. | 380 |
| $K_{g,i}$ | Grazing half-saturation constant for zooplankton $i$. | mmol m$^{-3}$ | |





## Acknowledgements


DT, HR, and EC were supported by grants from the Simons Foundation,

United States (Grant ID: 690671) and the NSF, United States (OCE- 2023680). SV was supported by the Trond

Mohn Foundation (Grant ID: TMS2018REK02).

## 390 Data Availability Statement

All observational data are from previously published literature.

## Code Availability Statement


Code used to generate model results are available in a publicly available online repository: https://github.com/werdna-spatial/GUD_closure

## Competing Interests


The declare that they have no competing interests.

## Autor Contributions

DT, SV, AW, and HR were involved in study design, DT and EC performed simulations, DT, EC, and AW performed model-data comparisons. DT, SV, and AW contributed to writing and editing.

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
