# Peer review of "Killing the predator: impacts of highest predator mortality on global-ocean ecosystem structure"

_Biogeosciences, 2023_

## Referee Comment (RC1)

Talmy et al. Biogeosci notes

General comments:

The authors use a simplified N-P-Z-D model to examine the influence of food web structure and mortality closure form on plankton community composition. They found that total phytoplankton biomass and plankton community composition were impacted by food web structure and the form of losses on the microzooplankton grazers. Results demonstrate the quadratic mortality is most consistent with the linear scaling of phytoplankton and microzooplankton biomasses seen in observations, and suggest that parallel food webs may be more appropriate for representing the coexistence of small and larger phytoplankton. This simplified experimental design allows for clear understanding of model structure influences ecosystem dynamics and of how much model complexity is required to capture observed biogeochemical properties. The results are bolstered by comparing model results to observed patterns (e.g. the linear scaling of phytoplankton and microzooplankton) rather than individual biomasses alone. I think this would be valuable to the BG readership.

The paper addresses relevant scientific questions within the scope of BG. The concepts, ideas, and tools are not necessarily novel, but their implementation and the comparison with recent observational datasets are. The scientific methods and assumptions are valid and clearly outlined, the results sufficient to support the interpretations and conclusions, and substantial conclusions are reached. The description of experiments and calculations are not sufficiently complete and precise to allow their reproduction by fellow scientists (traceability of results). Additionally, the authors do not give enough credit to related work, though they do clearly indicate their own new contribution. I do not think that the title clearly reflect the contents of the paper, specifically the use of the term "top predator." The abstract provides a concise and complete summary, the overall presentation is well structured and clear, and the language is fluent and precise. All mathematical formulae, symbols, abbreviations, and units are correctly defined and used, though parameter values, which I think are necessary, are not given. As such, the amount and quality of supplementary material is inappropriate. However, some formulae in appendix can be abbreviated. Finally, the number and quality of references are appropriate.

Specific comments:

1. Title and terminology
I would argue this term is misleading because microzooplankton are rarely considered top predators, even if they are the end of the food chain as modeled in this paper and others. Most people think of a top predator as a carnivore with few or no predators themselves; e.g. L64-65: "higher predation on the top predator" is counterintuitive. "Highest predator," (L136) "highest predator modeled/represented," or "terminal predator" may be better.

2. Take-away message

2a. L19: "importance of parameterizing" Density-dependent mortality on the terminal predator has been included in these models for decades. The closure term was always chosen to get

plankton dynamics "correct" (near those observed). I think that L41 may be the more relevant significance of this paper, i.e. understanding "how much model complexity is required to capture biogeochemically relevant properties." Specifically, the underlying drivers of linear scaling between microbial predators and prey are expounded.

2b. L44-46, L50-52: I really like the use of relationships/patterns found in observations to inform modeling or to assess models. I think this is an underutilized tool in the field. Such comparisons have been made by Luo et al. (2022) and Petrik et al. (2022), albeit chlorophyll instead of phytoplankton biomass, but they should be cited.

3. Comparisons with ESMs

3a. L66-67: I think this statement is false. BGC model developers put a significant amount of effort into calibrating mortality equation structure and parameterization. Often they are trying to capture simple metrics like total NPP or the spatial patterns in the global distribution of chlorophyll, but others go beyond that by calibrating against plankton community composition, Z:P spatial patterns, seasonal plankton biomass patterns, z-ratio, e-ratio, etc. (see Aumont & Bopp 2006, Aumont et al. 2015, Stock & Dunne 2010, Stock et al. 2014, Yool et al. 2013, 2021). Following on this, the authors have shown how phytoplankton community composition can vary based on food web structure and zooplankton mortality form. Do these choices also influence other important BGC quantities like NPP, export production, and the amount of secondary production available for zooplankton consumers?

3b. L95-96, L105, L264: These statements suggest that both parallel and diamond food web models are very prominent in the OBCG component of ESMs. How common is either of these? And is one more common than the other? It would be nice to have more quantitative idea of this. How many global ESMs only have microzooplankton (not one zooplankton that represents both)? And how many of them have parallel vs. diamond feeding? You could focus on just those that participated in CMIP5 or CMIP6. For example in Rohr et al. (2023): 3 with only 1P, 1Z (HAMOCC, CMOC, WOMBAT); 2 diamond (OECO, MARBL); 2 parallel (CanOE, COBALT); 3 hybrid (MEDUSA, PICES, BFM). Or Kearney et al. (2021): 1 with 0P, 0Z (BLING); 4 with 1P, 1Z (CMOC, WOMBAT, MRI, HAMOCC); 3 diamond (OECO, MARBL, NOBM); 2 parallel (CanOE, COBALT); 2 hybrid (MEDUSA, PICES). I don't expect the authors to list all these details in the paper, but suggest at their prevalence (e.g. ~30% of CMIP5 ESMs).

4. Missing information on experiment design and parameters.

4a. L109-110: Are the growth and grazing parameters different by size? Where can I find these?

4b. L114: What is the sensitivity of the results to these assumed sizes? I would argue that when ESMs use only one zooplankton type, it is supposed to represent both micro and mesozoo. Similarly, when there are two zoo types, one is micro and one is meso, and the meso preys on the micro, which is missing from the parallel food web here. Also, the "large phytoplankton" here is barely the size of the diatoms in ESMs (10-100 um, "microplankton"), while both the small and large microzooplankton are also at the low end of the microzooplankton (10-200 um in ESMs, 2-200 um in Sieburth et al. 1978), and when there is only one zooplankton group in ESMs it tends

to encompass everything 10-2,000 um. I do not mean to suggest that this study is without value for that reason, but the comparison to global ESMs used for climate change studies is less direct.

4c. L127-128: Is the linear mortality term included in both versions? Or is it either linear or quadratic mortality? It is rare for ESMs to have either linear or non-linear mortality terms on their zooplankton. The linear loss term in these models usually accounts for metabolic losses while the non-linear term accounts for higher predator mortality (see Kearney et al. 2021, Petrik et al. 2022). This is alluded to on L276-279.

4d. Table 1: Did $\delta_Z$ and $\delta_{ZZ}$ have the same values in the parallel and diamond food webs? Or all the parameters for that matter?

4e. Figure 2: Are the arrows meant to line up with (phyto→PON) and (zoop→DOP) or is that just a coincidence?

4f. Appendix Equation A34: What are $g_{max}$ and $K_{1/2}$ for the two microzoo types? Are all parameters held constant between the model version?

4g. Table A3: The parameter values pertaining to zooplankton mortality and grazing should be given here since that is the focus of the paper. Also, I could not find these parameters on the website listed. Please give the filename and folder/directory with these values.

4h. Table A3. "The large phytoplankton has a faster maximal growth rate and higher nutrient half-saturation constants than the small phytoplankton, representative of differences in growth rate between a eukaryotic algae and a cyanobacteria, respectively (Litchman et al. 2007; Ward et al. 2012)." Is this true of the microzooplankton when there are two size classes?

5. Other

5a. L102-104: How do the model results presented in this paper compare to observations in the Pacific (e.g. Follett et al. 2022)? This seems relevant as much of the early literature cited on linear vs. quadratic mortality were trying to capture the difference in the seasonal cycle between the N Pacific and N Atlantic.

5b. Figure 3 and results: Why just surface biomass? Aren't plankton distributed throughout the euphotic layer? Subsurface biomass could show different patterns. I suggest analyzing and depicting depth-integrated biomass.

5c. Figure 3 and results: Could you please also show the microzooplankton biomass across the different models? I am curious how much it showed similar patterns to the phytoplankton, which would be expected for the quadratic closure, but not for the linear. This would help understanding the spatial differences that can't be seen in Fig 5.

5d. L230: Linear mortality resulting in oscillations is known behavior. Please mention in the discussion and refer to the relevant literature (e.g. Steele & Henderson 1992, Fasham 1995, Edwards & Brindley 1999, Edwards & Yool 2000).

5e. Figure 6 and results: The statement that "Linear losses on the microzooplankton predict cyclic behavior in the predator-prey relationship that are inconsistent with observations" is not accurate. The slope of (a) and (b) are inconsistent with the global pattern, which are snapshots in time, but it is possible that a similar seasonal pattern exists in the observations. You would need to show those. Given the wealth of data in the English Channel from the CPR, I assume this is possible.

Technical corrections:

Appendix Equations A1-A10: These are all the same form and don't all need to be shown. You could simply use an "X" or something to denote the nutrient on DOM. Also, A1 has NO3 in the equation, but should be NH4 throughout.

Appendix Equations A11-A14: Again, could just use one example of POM.

References cited:
Aumont, O., Ethé, C., Tagliabue, A., Bopp, L., & Gehlen, M. (2015). PISCES-v2: An ocean biogeochemical model for carbon and ecosystem studies. Geoscientific Model Development, 8, 2465–2513. https://doi.org/10.5194/gmd-8-2465-2015

Aumont, O., Maury, O., Lefort, S., & Bopp, L. (2018). Evaluating the potential impacts of the diurnal vertical migration by marine organisms on marine biogeochemistry. Global Biogeochemical Cycles, 32(11), 1622–1643. https://doi.org/10.1029/2018GB005886

Edwards, A. M., & Yool, A. (2000). The role of higher predation in plankton population models. Journal of Plankton Research, 22(6), 1085-1112.

Fasham,M.J.R. (1995) Variations in the seasonal cycle of biological production in subarctic oceans: A model sensitivity analysis. Deep-Sea Res. I, 42, 1111–1149.

Kearney, K. A., Bograd, S. J., Drenkard, E., Gomez, F. A., Haltuch, M., Hermann, A. J., et al. (2021). Using global-scale Earth system models for regional fisheries applications. Frontiers in Marine Science, 1121. https://doi.org/10.3389/fmars.2021.622206

Luo, J. Y., Stock, C. A., Henschke, N., Dunne, J. P., & O'Brien, T. D. (2022). Global ecological and biogeochemical impacts of pelagic tunicates. Progress in Oceanography, 205, 102822. https://doi.org/10.1016/j.pocean.2022.102822

Petrik, C. M., Luo, J. Y., Heneghan, R. F., Everett, J. D., Harrison, C. S., & Richardson, A. J. (2022). Assessment and constraint of mesozooplankton in CMIP6 Earth system models. Global Biogeochemical Cycles, 36, e2022GB007367. https://doi. org/10.1029/2022GB007367

Sieburth, J. M., Smetacek, V., & Lenz, J. (1978). Pelagic ecosystem structure: Heterotrophic compartments of the plankton and their relationship to plankton size fractions 1. Limnology and oceanography, 23(6), 1256-1263.

Stock, C., & Dunne, J. (2010). Controls on the ratio of mesozooplankton production to primary production in marine ecosystems. Deep Sea Research Part I: Oceanographic Research Papers, 57(1), 95-112.

Stock, C. A., Dunne, J. P., & John, J. G. (2014). Global-scale carbon and energy flows through the marine planktonic food web: An analysis with
a coupled physical-biological model. Progress in Oceanography, 120, 1–28. https://doi.org/10.1016/j.pocean.2013.07.001

Yool, A., Popova, E. E., & Anderson, T. R. (2011). Medusa-1.0: A new intermediate complexity plankton ecosystem model for the global domain. Geoscientific Model Development, 4(2), 381–417. https://doi.org/10.5194/gmd-4-381-2011

Yool, A., Popova, E. E., & Anderson, T. R. (2013). MEDUSA-2.0: An intermediate complexity biogeochemical model of the marine carbon cycle for climate change and ocean acidification studies. Geoscientific Model Development, 6(5), 1767–1811. https://doi.org/10.5194/ gmd-6-1767-2013

---

## Author Comment (AC1)

General comments:

Talmy et al. Biogeosci notes

The authors use a simplified N-P-Z-D model to examine the influence of food web structure and mortality closure form on plankton community composition. They found that total phytoplankton biomass and plankton community composition were impacted by food web structure and the form of losses on the microzooplankton grazers. Results demonstrate the quadratic mortality is most consistent with the linear scaling of phytoplankton and microzooplankton biomasses seen in observations and suggest that parallel food webs may be more appropriate for representing the coexistence of small and larger phytoplankton. This simplified experimental design allows for clear understanding of model structure influences ecosystem dynamics and of how much model complexity is required to capture observed biogeochemical properties. The results are bolstered by comparing model results to observed patterns (e.g. the linear scaling of phytoplankton and microzooplankton) rather than individual biomasses alone. I think this would be valuable to the BG readership.

The paper addresses relevant scientific questions within the scope of BG. The concepts, ideas, and tools are not necessarily novel, but their implementation and the comparison with recent observational datasets are. The scientific methods and assumptions are valid and clearly outlined, the results sufficient to support the interpretations and conclusions, and substantial conclusions are reached. The description of experiments and calculations are not sufficiently complete and precise to allow their reproduction by fellow scientists (traceability of results). Additionally, the authors do not give enough credit to related work, though they do clearly indicate their own new contribution. I do not think that the title clearly reflect the contents of the paper, specifically the use of the term "top predator." The abstract provides a concise and complete summary, the overall presentation is well structured and clear, and the language is fluent and precise. All mathematical formulae, symbols, abbreviations, and units are correctly defined and used, though parameter values, which I think are necessary, are not given. As such, the amount and quality of supplementary material is inappropriate. However, some formulae in appendix can be abbreviated. Finally, the number and quality of references are appropriate.

We are grateful for this careful review and constructive criticism of our work. We are preparing a revised text, and in it we will provide an expanded supplemental information with appropriate sensitivity studies and model output, specific parameter values, and more detailed representation of our allometric size-rules. We will also change the title in line with the reviewer's suggestion. We provide more details of our planned revisions below.

Specific comments:

1. Title and terminology
I would argue this term is misleading because microzooplankton are rarely considered top predators, even if they are the end of the food chain as modeled in this paper and others. Most people think of a top predator as a carnivore with few or no predators themselves; e.g. L64-65: "higher predation on the top predator" is counterintuitive. "Highest predator," (L136) "highest predator modeled/represented," or "terminal predator" may be better.

We plan to change the title to: "*Killing the predator: impact of highest predator mortality on global ocean ecosystem structure*"

2. Take-away message

2a. L19: "importance of parameterizing" Density-dependent mortality on the terminal predator has been included in these models for decades. The closure term was always chosen

to get plankton dynamics "correct" (near those observed). I think that L41 may be the more relevant significance of this paper, i.e. understanding "how much model complexity is required to capture biogeochemically relevant properties." Specifically, the underlying drivers of linear scaling between microbial predators and prey are expounded.

This is an interesting point. We plan to rewrite the abstract to focus on the closure as an explanation of linear predator-prey scaling in models with minimal complexity.

2b. L44-46, L50-52: I really like the use of relationships/patterns found in observations to inform modeling or to assess models. I think this is an underutilized tool in the field. Such comparisons have been made by Luo et al. (2022) and Petrik et al. (2022), albeit chlorophyll instead of phytoplankton biomass, but they should be cited.

Thank you – we will be sure to add these important citations

3. Comparisons with ESMs

3a. L66-67: I think this statement is false. BGC model developers put a significant amount of effort into calibrating mortality equation structure and parameterization. Often they are trying to capture simple metrics like total NPP or the spatial patterns in the global distribution of chlorophyll, but others go beyond that by calibrating against plankton community composition, Z:P spatial patterns, seasonal plankton biomass patterns, z-ratio, e-ratio, etc. (see Aumont & Bopp 2006, Aumont et al. 2015, Stock & Dunne 2010, Stock et al. 2014, Yool et al. 2013, 2021). Following on this, the authors have shown how phytoplankton community composition can vary based on food web structure and zooplankton mortality form. Do these choices also influence other important BGC quantities like NPP, export production, and the amount of secondary production available for zooplankton consumers?

We will be sure to cite these important works in our revisions and explain how our work builds on the foundation they have set. We are also in the process of outputting additional BGC quantities. We have NPP, zooplankton biomass and carbon export. We are a little unclear what is meant by the 'amount of secondary production available for zooplankton consumers'. We have tentatively assumed that this refers to the quadratic closure (as this parameterizes losses from microzooplankton to higher trophic levels, e.g. mesozooplankton). However, we would welcome clarification as we prepare our revisions.

3b. L95-96, L105, L264: These statements suggest that both parallel and diamond food web models are very prominent in the OBCG component of ESMs. How common is either of these? And is one more common than the other? It would be nice to have more quantitative idea of this. How many global ESMs only have microzooplankton (not one zooplankton that represents both)? And how many of them have parallel vs. diamond feeding? You could focus on just those that participated in CMIP5 or CMIP6. For example in Rohr et al. (2023): 3 with only 1P, 1Z (HAMOCC, CMOC, WOMBAT); 2 diamond (OECO, MARBL); 2 parallel (CanOE, COBALT); 3 hybrid (MEDUSA, PICES, BFM). Or Kearney et al. (2021): 1 with 0P, 0Z (BLING); 4 with 1P, 1Z (CMOC, WOMBAT, MRI, HAMOCC); 3 diamond (OECO, MARBL, NOBM); 2 parallel (CanOE, COBALT); 2 hybrid (MEDUSA, PICES). I don't expect the authors to list all these details in the paper, but suggest at their prevalence (e.g. ~30% of CMIP5 ESMs).

This is a helpful suggestion. We plan to discuss these differences in food-web structure between Earth System Models in our revisions.

4. Missing information on experiment design and parameters.
4a. L109-110: Are the growth and grazing parameters different by size? Where can I find these?

*Yes, both growth and grazing rates are allometric. The rules used are specified in Ward et al. (2012), and we will provide a table with them in our revisions – apologies for this omission.*

4b. L114: What is the sensitivity of the results to these assumed sizes? I would argue that when ESMs use only one zooplankton type, it is supposed to represent both micro and mesozoo. Similarly, when there are two zoo types, one is micro and one is meso, and the meso preys on the micro, which is missing from the parallel food web here. Also, the "large phytoplankton" here is barely the size of the diatoms in ESMs (10-100 um, "microplankton"), while both the small and large microzooplankton are also at the low end of the microzooplankton (10-200 um in ESMs, 2- 200 um in Sieburth et al. 1978), and when there is only one zooplankton group in ESMs it tends to encompass everything 10-2,000 um. I do not mean to suggest that this study is without value for that reason, but the comparison to global ESMs used for climate change studies is less direct.

*Interestingly, there seems to be a great deal of variation regarding what is being described in ESMs. From Rohr et al., 2023: "it is concerning that some models imply something statistically similar to an ocean filled entirely with very slow-grazing meroplankton larvae (MEDUSA2.1, OECOv2) and others an ocean filled entirely with very rapidly-grazing ciliates (MARBL and CMOC)."*

*We are currently conducting sensitivities to assumed phytoplankton and zooplankton size classes, that we plan to include in our revisions. We will be sure to interpret our output in the context of the existing range of organism sizes represented in ESMs.*

4c. L127-128: Is the linear mortality term included in both versions? Or is it either linear or quadratic mortality? It is rare for ESMs to have either linear or non-linear mortality terms on their zooplankton. The linear loss term in these models usually accounts for metabolic losses while the non-linear term accounts for higher predator mortality (see Kearney et al. 2021, Petrik et al. 2022). This is alluded to on L276-279.

*Great question, we did have the linear turned on with the quadratic but have since conducted a sensitivity where we switch it off. Turning off the linear in the quadratic case appears to have negligible impact on the properties we report, indicating that quadratic closure may have an outsized impact on ecosystem properties.*

4d. Table 1: Did $\delta_Z$ and $\delta_{ZZ}$ have the same values in the parallel and diamond food webs? Or all the parameters for that matter?

*Yes, we will clarify this in our revisions.*

4e. Figure 2: Are the arrows meant to line up with (phyto→PON) and (zoop→DOP) or is that just a coincidence?

*This is a coincidence – we will clarify this in our revisions.*

4f. Appendix Equation A34: What are $g_{max}$ and $K_{1/2}$ for the two microzoo types? Are all parameters held constant between the model version?

*Yes, all parameters are held constant. We will report all parameter values in our revised manuscript*

4g. Table A3: The parameter values pertaining to zooplankton mortality and grazing should be given here since that is the focus of the paper. Also, I could not find these parameters on the website listed. Please give the filename and folder/directory with these values.

*We will add these values to the manuscript and also to our GitHub repository.*

4h. Table A3. "The large phytoplankton has a faster maximal growth rate and higher nutrient half-saturation constants than the small phytoplankton, representative of differences in growth rate between a eukaryotic algae and a cyanobacteria, respectively (Litchman et al. 2007; Ward et al. 2012)." Is this true of the microzooplankton when there are two size classes?

Yes, the larger zooplankton have slower maximal growth rates consistent with allometric scaling. We will report this in our revisions.

5. Other

5a. L102-104: How do the model results presented in this paper compare to observations in the Pacific (e.g. Follett et al. 2022)? This seems relevant as much of the early literature cited on linear vs. quadratic mortality were trying to capture the difference in the seasonal cycle between the N Pacific and N Atlantic.

This is an interesting question. We are currently trying to identify a dataset in the Pacific with microzooplankton and phytoplankton carbon density. The Hawaii Ocean Time Series appears to only have mesozooplankton (>200 micron). Although some datasets include microzooplankton carbon (e.g. https://doi.pangaea.de/10.1594/PANGAEA.779970), it is unclear to us if these have coincident phytoplankton carbon density associated with them. We will continue to search for appropriate data in the Pacific.

5b. Figure 3 and results: Why just surface biomass? Aren't plankton distributed throughout the euphotic layer? Subsurface biomass could show different patterns. I suggest analyzing and depicting depth-integrated biomass.

We have already prepared depth integrated figures and will include these in our revisions

5c. Figure 3 and results: Could you please also show the microzooplankton biomass across the different models? I am curious how much it showed similar patterns to the phytoplankton, which would be expected for the quadratic closure, but not for the linear. This would help understanding the spatial differences that can't be seen in Fig 5.

We have zooplankton biomasses that we will provide. Note that earlier versions of the manuscript had the figure below, showing how carbon accumulates in the different groups. There are interesting differences in the way carbon builds up, with linear closure leading to far tighter constraint on the phytoplankton, and greater accumulation in the zooplankton (top row). We omitted this figure as some readers find it confusing. The maps we are preparing hopefully will depict this in a more intuitive way.

[Figure]
* * *
5d. L230: Linear mortality resulting in oscillations is known behavior. Please mention in the discussion and refer to the relevant literature (e.g. Steele & Henderson 1992, Fasham 1995, Edwards & Brindley 1999, Edwards & Yool 2000).

We will add these citations.

5e. Figure 6 and results: The statement that "Linear losses on the microzooplankton predict cyclic behavior in the predator-prey relationship that are inconsistent with observations" is not accurate. The slope of (a) and (b) are inconsistent with the global pattern, which are snapshots in time, but it is possible that a similar seasonal pattern exists in the observations. You would need to show those. Given the wealth of data in the English Channel from the CPR, I assume this is possible.

We will show the seasonal pattern at L4 – it indicates that phyto and zoo biomass is very highly correlated.

Technical corrections:

Appendix Equations A1-A10: These are all the same form and don't all need to be shown. You could simply use an "X" or something to denote the nutrient on DOM. Also, A1 has NO3 in the equation, but should be NH4 throughout.

Appendix Equations A11-A14: Again, could just use one example of POM.

We'll be sure to fix these.

References cited:

Aumont, O., Ethé, C., Tagliabue, A., Bopp, L., & Gehlen, M. (2015). PISCES-v2: An ocean biogeochemical model for carbon and ecosystem studies. Geoscientific Model Development, 8, 2465–2513. https://doi.org/10.5194/gmd-8-2465-2015

Aumont, O., Maury, O., Lefort, S., & Bopp, L. (2018). Evaluating the potential impacts of the diurnal vertical migration by marine organisms on marine biogeochemistry. Global Biogeochemical Cycles, 32(11), 1622–1643. https://doi.org/10.1029/2018GB005886

Edwards, A. M., & Yool, A. (2000). The role of higher predation in plankton population models. Journal of Plankton Research, 22(6), 1085-1112.

Fasham,M.J.R. (1995) Variations in the seasonal cycle of biological production in subarctic oceans: A model sensitivity analysis. Deep-Sea Res. I, 42, 1111–1149.

Kearney, K. A., Bograd, S. J., Drenkard, E., Gomez, F. A., Haltuch, M., Hermann, A. J., et al. (2021). Using global-scale Earth system models for regional fisheries applications. Frontiers in Marine Science, 1121. https://doi.org/10.3389/fmars.2021.622206

Luo, J. Y., Stock, C. A., Henschke, N., Dunne, J. P., & O'Brien, T. D. (2022). Global ecological and biogeochemical impacts of pelagic tunicates. Progress in Oceanography, 205, 102822. https://doi.org/10.1016/j.pocean.2022.102822

Petrik, C. M., Luo, J. Y., Heneghan, R. F., Everett, J. D., Harrison, C. S., & Richardson, A. J. (2022). Assessment and constraint of mesozooplankton in CMIP6 Earth system models. Global Biogeochemical Cycles, 36, e2022GB007367. https://doi.org/10.1029/2022GB007367

Sieburth, J. M., Smetacek, V., & Lenz, J. (1978). Pelagic ecosystem structure: Heterotrophic compartments of the plankton and their relationship to plankton size fractions 1. Limnology and oceanography, 23(6), 1256-1263.

Stock, C., & Dunne, J. (2010). Controls on the ratio of mesozooplankton production to primary production in marine ecosystems. Deep Sea Research Part I: Oceanographic Research Papers, 57(1), 95-112.

Stock, C. A., Dunne, J. P., & John, J. G. (2014). Global-scale carbon and energy flows through the marine planktonic food web: An analysis with a coupled physical-biological model. Progress in Oceanography, 120, 1–28. https://doi.org/10.1016/j.pocean.2013.07.001

Yool, A., Popova, E. E., & Anderson, T. R. (2011). Medusa-1.0: A new intermediate complexity plankton ecosystem model for the global domain. Geoscientific Model Development, 4(2), 381– 417. https://doi.org/10.5194/gmd-4-381-2011

Yool, A., Popova, E. E., & Anderson, T. R. (2013). MEDUSA-2.0: An intermediate complexity biogeochemical model of the marine carbon cycle for climate change and ocean acidification studies. Geoscientific Model Development, 6(5), 1767–1811. https://doi.org/10.5194/ gmd-6- 1767-2013

---

## Author Comment (AC2)

Talmy et al. "Killing the predator: impacts of top-predator mortality on global-ocean ecosystem structure"

This is a nice, concise paper analyzing the effects of two variants of a plankton food web structure, with two types of losses (linear and quadratic) for the top predator in the modeled food web. Here, the predators are microzooplankton, but I believe the results are extensible to a case where there is one microzooplankton and one mesozooplankton. In Talmy et al., the authors show that the "diamond" food web structure, where one zooplankton feeds on two phytoplankton, results in a marine ecosystem with less dynamic range between the gyres and the poles in phytoplankton carbon, as well as less co-existence in community composition. Simultaneously, the quadratic losses in the top predators results in a phytoplankton to zooplankton biomass relationship that better represents recent observations. I believe This paper is written in a clear and consise way, but one major criticism is its lack of robust engagement in other mechanisms that may contribute to variations in community co-existence, Z:P ratio, and phytoplankton carbon. Further, while the authors do cite other publications (Ward et al. 2012, Dutkiewicz et al. 2020) that list parameter values, it was very difficult to evaluate the performance of the model without a list of parameter values.

Thank you for these constructive comments. We appreciate the suggestions to better engaging with existing literature and to provide parameter values. We are in the process of running some targeted simulations to investigate a subset of these interesting questions. We specify these details below.

Therefore, my recommendations are:

- List out in the appendix all the ecosystem-relevant parameter values used in the model

  We are preparing a detailed set of tables reporting the allometric rules and parameter choices, that we will be sure to include in our revised manuscript.

- Can the authors engage a bit more robustly in the discussion, other strategies that modelers use to modify zooplankton grazing in a diamond food web structure that may also result in improvements in the two main metrics (dynamic range between gyres and poles in phytoplankton carbon, community co-existence):

  1. For example, many biogeochemical models utilize either different maximum grazing rates for zooplankton depending on the prey type, OR use a prey selectivity factor to modulate grazing. For the former, BEC/MARBL (Moore et al. 2004, Long et al. 2021) uses different maximum grazing rates for a single "adaptive" zooplankton to mimic the effect of multiple zooplankton in a single zooplankton class. For the latter, there are multiple examples of this strategy within the biogeochemical models, e.g., PISCES (Aumont et al. 2015), and COBALT (Stock et al. 2014), though those two models also have multiple zooplankton types so it may be slightly harder to compare with a single zooplankton type. However, mathematically, the effect of both these strategies would be similar.

     In the parallel food chain, the larger zooplankton feeding on the larger phytoplankton has a lower maximal growth rate. In response to reviewer 1's comments, we are conducting sensitivity studies to explore whether our solutions are modified drastically depending on grazer growth rates. This does not directly address the reviewer's comments but is a related concept.

We will also be sure to add text to our Discussion raising the potential to explore the impact of adaptive predation on predator-prey scaling as an exciting future direction, citing the literature mentioned here.

2. On grazing, values for the zooplankton maximum grazing rates and the grazing half-saturation constant are amongst the least well constrained parameters in food web models, and variations in these parameters have an enormous impact. Rohr et al. 2022 (Progress in Oceanography) shows this quite nicely in a robust analysis, along with evaluating differences in the grazing functional form itself (Holling type II or type III functional responses). It would be nice if the authors could engage a bit more in the discussion regarding whether modelers would be able to compensate for the lack of a second zooplankton (e.g., in the diamond food web model) by modulating maximum grazing rates and grazing half-saturation constants.

It is interesting that Rohr at al. (2022) rely on a Holling Type-III functional response to stabilize oscilliatory dynamics, in a system with linear zooplankton mortality. Out of curiosity, we plan to run some simulations with a Holling Type-III response and linear closure, to see if this also can explain linear predator-prey scaling. Regarding mimicking a second zooplankton in the diamond food-web, we will be sure to cite the relevant literature, and point to this as an exciting avenue for future study.

3. Lastly – prey switching is a major issue that is only mentioned in passing in the discussion. It would be nice to see a more robust discussion – do the authors think that modifications in the switching form would result in substantial changes in the modeled ecosystem, and why? There are a lot of approaches towards switching, as laid out extensively in Gentleman et al. (2003), but models typically use just one or two forms (e.g., Stock et al. 2008 Journal of Marine Systems has addressed this quite nicely in a simple system). In my opinion, a more than cursory treatment of this topic would be important in this paper.

We recognize that we can engage more fully with the large body of literature describing different approaches to prey switching and appreciate the suggestion from this reviewer to do so. Out of curiosity, we have begun running some simulations contrasting two well-known forms of switching - passive vs. active -  following guidelines of Gentleman et al. (2003) and Vallina et al. (2014). Early results suggest that active switching promotes coexistence among small and large phytoplankton in the diamond food-web, but does not strongly impact predator-prey scaling, total planktonic carbon, and carbon export. We will be sure to discuss these findings in the context of the wider range of options available for prey switching, and the potential for future studies exploring the impacts of these on predator-prey biomass scaling relationships.

4. Other parameters that may additionally modulate phytoplankton carbon in food web models that aren't addressed include the fraction of phytoplankton and zooplankton losses that go to dissolved organic matter vs. particulate organic matter, which may influence the recycling rate and strength of the microbial loop. Lastly, variations in the relative nutrient uptake rate of the different phytoplankton may also result in more or less differences in the phytoplankton carbon between the gyres and the poles.

We are conducting sensitivities to phytoplankton size, which will modify nutrient affinities (through allometry). Also, we have conducted sensitives to the POM-DOM partitioning (referred to in our model as an 'export fraction'). Results show that, when modified over a reasonable range, the average predator-prey scaling is sensitive to this number, but the scaling relationship is not. If there is space in our revisions, we will include these results.

Other than these points, I found the manuscript written quite clearly, with compelling figures and nice presentation. With a more robust discussion addressing a range of these additional points listed above, this manuscript would make a nice addition to the literature.

We thank this reviewer for the careful reading of our manuscript, and their helpful suggestions. We anticipate our manuscript will be much improved as a result.

---

## Author Response (AR1)

Dear Biogeosciences Editors,

Please find enclosed our revisions along with point-by-point responses to all reviewer comments. We now include an exhaustive list of all parameter values in our online code repository, and the most relevant ecosystem parameters are provided in the main text of the manuscript. We have also conducted simulations testing the sensitivity of our main findings to different assumptions about plankton size and microzooplankton feeding mechanisms. Note that, since our last submission, we updated our code to use a more recent version of the Darwin ecosystem model. None of our main conclusions are qualitatively impacted, but several of the figures have minor aesthetic differences due to small changes in parameter values.

We have also now included more discussion of prior work as it relates to our main conclusions. Overall, we are very grateful to both reviewers for providing thoughtful and constructive comments that we feel have led to deeper understanding of our model simulations that has improved the manuscript.

Kind regards,

David Talmy, Eric Carr, Selina Våge, Harshana Rajakaruna, and Anne Willem Omta

**Reviewer 1**

General comments:

Talmy et al. Biogeosci notes

The authors use a simplified N-P-Z-D model to examine the influence of food web structure and mortality closure form on plankton community composition. They found that total phytoplankton biomass and plankton community composition were impacted by food web structure and the form of losses on the microzooplankton grazers. Results demonstrate the quadratic mortality is most consistent with the linear scaling of phytoplankton and microzooplankton biomasses seen in observations and suggest that parallel food webs may be more appropriate for representing the coexistence of small and larger phytoplankton. This simplified experimental design allows for clear understanding of model structure influences ecosystem dynamics and of how much model complexity is required to capture observed biogeochemical properties. The results are bolstered by comparing model results to observed patterns (e.g. the linear scaling of phytoplankton and microzooplankton) rather than individual biomasses alone. I think this would be valuable to the BG readership.

The paper addresses relevant scientific questions within the scope of BG. The concepts, ideas, and tools are not necessarily novel, but their implementation and the comparison with recent observational datasets are. The scientific methods and assumptions are valid and clearly outlined, the results sufficient to support the interpretations and conclusions, and substantial conclusions are reached. The description of experiments and calculations are not sufficiently complete and precise to allow their reproduction by fellow scientists (traceability of results). Additionally, the authors do not give enough credit to related work, though they do clearly indicate their own new contribution. I do not think that the title clearly reflect the contents of the paper, specifically the use of the term "top predator." The abstract provides a concise and complete summary, the overall presentation is well structured and clear, and the language is fluent and precise. All mathematical formulae, symbols, abbreviations, and units are correctly defined and used, though parameter values, which I think are necessary, are not

given. As such, the amount and quality of supplementary material is inappropriate. However, some formulae in appendix can be abbreviated. Finally, the number and quality of references are appropriate.

*Thank you for these constructive remarks. We have now provided an exhaustive list of all model parameter values in our online code repository, and a subset of these is provided in the main text. We have also provided results showing additional biogeochemical model predictions in a supplementary file. We respond to each specific comment below.*

Specific comments:

1. Title and terminology
I would argue this term is misleading because microzooplankton are rarely considered top predators, even if they are the end of the food chain as modeled in this paper and others. Most people think of a top predator as a carnivore with few or no predators themselves; e.g. L64-65: "higher predation on the top predator" is counterintuitive. "Highest predator," (L136) "highest predator modeled/represented," or "terminal predator" may be better.

*We have changed the title to: "Killing the predator: impacts of highest predator mortality on global-ocean ecosystem structure" and used this terminology throughout the manuscript.*

2. Take-away message

2a. L19: "importance of parameterizing" Density-dependent mortality on the terminal predator has been included in these models for decades. The closure term was always chosen to get plankton dynamics "correct" (near those observed). I think that L41 may be the more relevant significance of this paper, i.e. understanding "how much model complexity is required to capture biogeochemically relevant properties." Specifically, the underlying drivers of linear scaling between microbial predators and prey are expounded.

*We have altered the abstract to emphasize that these biomass scaling relationships can be reproduced with models of low complexity (Lines 20-21, new text underlined):*

*"Our findings point to the importance of parameterizing mortality of the highest predator for simple food web models to recapitulate trophic structure in the global ocean"*

*We are reluctant to claim that we can be sure how much complexity is required to capture biogeochemistry, as our study is limited to understanding just one small property of these systems, namely the biomass scaling relationship between microzooplankton predators and their phytoplankton prey.*

2b. L44-46, L50-52: I really like the use of relationships/patterns found in observations to inform modeling or to assess models. I think this is an underutilized tool in the field. Such comparisons have been made by Luo et al. (2022) and Petrik et al. (2022), albeit chlorophyll instead of phytoplankton biomass, but they should be cited.

*We have added these citations (Line 45)*

3. Comparisons with ESMs

3a. L66-67: I think this statement is false. BGC model developers put a significant amount of effort into calibrating mortality equation structure and parameterization. Often they are trying to capture simple metrics like total NPP or the spatial patterns in the global distribution of chlorophyll, but others go beyond that by calibrating against plankton community composition, Z:P spatial patterns, seasonal plankton biomass patterns, z-ratio, e-ratio, etc. (see Aumont & Bopp 2006, Aumont et al. 2015, Stock & Dunne 2010, Stock et al. 2014, Yool et al. 2013, 2021). Following on this, the authors have shown how phytoplankton community composition can vary based on food web structure and zooplankton mortality

form. Do these choices also influence other important BGC quantities like NPP, export production, and the amount of secondary production available for zooplankton consumers?

We have modified this statement (Lines 67-70):

"*The assumptions made here profoundly influence biogeochemical properties such as primary production and chlorophyll distribution (Aumont and Bopp 2006; Stock and Dunne 2010; Yool et al. 2013; Stock et al. 2014; Aumont et al. 2015). However, it is unclear if and how their effects are dependent on choices made about food web structure.*"

And added maps showing depth integrated primary production, carbon export, and secondary production to the supplementary information (Figs S3-S5)

3b. L95-96, L105, L264: These statements suggest that both parallel and diamond food web models are very prominent in the OBCG component of ESMs. How common is either of these? And is one more common than the other? It would be nice to have more quantitative idea of this. How many global ESMs only have microzooplankton (not one zooplankton that represents both)? And how many of them have parallel vs. diamond feeding? You could focus on just those that participated in CMIP5 or CMIP6. For example in Rohr et al. (2023): 3 with only 1P, 1Z (HAMOCC, CMOC, WOMBAT); 2 diamond (OECO, MARBL); 2 parallel (CanOE, COBALT); 3 hybrid (MEDUSA, PICES, BFM). Or Kearney et al. (2021): 1 with 0P, 0Z (BLING); 4 with 1P, 1Z (CMOC, WOMBAT, MRI, HAMOCC); 3 diamond (OECO, MARBL, NOBM); 2 parallel (CanOE, COBALT); 2 hybrid (MEDUSA, PICES). I don't expect the authors to list all these details in the paper, but suggest at their prevalence (e.g. ~30% of CMIP5 ESMs).

Interestingly, in most cases where there is just one zooplankton, it is usually 'adaptive', intended to represent both micro and meso-zooplankton (see also reviewer 2 comments on this subject). Also, many of the CMIP6 models have elements of both parallel AND diamond feeding (e.g. COBALT and MEDUSA). Therefore, one could argue that in fact none of our models match exactly with any of the CMIP6 models. Nevertheless, parallel feeding and shared predation are components of many of these models, and by evaluating these model properties in isolation, we aspire to inform development of more complex models with other considerations. We have therefore taken this reviewers suggestion but identified all models evaluated by Rohr et al., with diamond (OECO-v2, MARBL, MEDUSA, PICES, BFM, COBALT) and/or parallel components (CanOE, MEDUSA, PICES, BFM, COBALT), and rephrased this section, e.g. (Line 99-101):

"*Furthermore, parallel feeding was a component of five of ten Earth system models that were part of the most recent Coupled Model Intercomparison project (CMIP6) evaluated by Rohr et al. (2023), making it a useful food-web structure to examine in a global ocean context.*"

We have also added a paragraph to the discussion recognizing the mismatch between our models and many ESMs (Line 318-326):

"*The food-web model structures assumed here are so simple that they exclude many mechanisms already considered in extant Earth system models. For example, many Earth system models contain representations of both microzooplankton and mesozooplankton. In some cases, these are explicitly represented with multiple state variables (Stock et al. 2014; Aumont et al. 2015). In others, a single 'adaptive' zooplankton class mimics the effects of micro- and meso-zooplankton by feeding differently on phytoplankton prey types (Moore et al. 2004; Long et al. 2021). Future studies may evaluate the impact of different closures in the context of these more sophisticated structures. Despite the simplicity of our models, we*

*anticipate that our central conclusions will hold in a more general setting. Specifically, assumptions made about highest predator mortality constrain biomass scaling relationships regardless of model predictions about community composition within a trophic level."*

4. Missing information on experiment design and parameters.
4a. L109-110: Are the growth and grazing parameters different by size? Where can I find these?

Yes, these vary according to allometric scaling relationships that are now provided in Table 3 of the main text

4b. L114: What is the sensitivity of the results to these assumed sizes? I would argue that when ESMs use only one zooplankton type, it is supposed to represent both micro and mesozoo. Similarly, when there are two zoo types, one is micro and one is meso, and the meso preys on the micro, which is missing from the parallel food web here. Also, the "large phytoplankton" here is barely the size of the diatoms in ESMs (10-100 um, "microplankton"), while both the small and large microzooplankton are also at the low end of the microzooplankton (10-200 um in ESMs, 2- 200 um in Sieburth et al. 1978), and when there is only one zooplankton group in ESMs it tends to encompass everything 10-2,000 um. I do not mean to suggest that this study is without value for that reason, but the comparison to global ESMs used for climate change studies is less direct.

Interestingly, there seems to be a great deal of variation regarding what is being described in ESMs. From Rohr et al., 2023: "*it is concerning that some models imply something statistically similar to an ocean filled entirely with very slow-grazing meroplankton larvae (MEDUSA2.1, OECOv2) and others an ocean filled entirely with very rapidly-grazing ciliates (MARBL and CMOC).*" Therefore, our modeled rates of grazing are likely to fall within the envelope of values simulated in ESMs. Nevertheless, we conducted sensitivity simulations modifying phytoplankton and zooplankton size (described in lines 174-179 and lines 282-283) and found that our primary conclusions regarding scaling relationships are insensitive to assumptions about size. These results are shown in Figure S6.

4c. L127-128: Is the linear mortality term included in both versions? Or is it either linear or quadratic mortality? It is rare for ESMs to have either linear or non-linear mortality terms on their zooplankton. The linear loss term in these models usually accounts for metabolic losses while the non-linear term accounts for higher predator mortality (see Kearney et al. 2021, Petrik et al. 2022). This is alluded to on L276-279.

Great question, we did have the linear turned on with the quadratic but have since conducted a sensitivity where we switch it off. Turning off the linear in the quadratic case appears to have negligible impact on the properties we report, indicating that quadratic closure may have an outsized impact on ecosystem properties.

We recognize that both sources of mortality are usually included in ESMs. However, many simpler ecological and biogeochemical models opt for linear closure (excluding quadratic) as these are often algebraically more manageable. We now clarify in the caption of table 2 that these values are switched to zero in simulations where they are not considered

4d. Table 1: Did $\delta_Z$ and $\delta_{ZZ}$ have the same values in the parallel and diamond food webs? Or all the parameters for that matter?

Yes, these are now provided in Table 2 and we clarify that these and all other parameters are held constant unless the entire term is neglected

4e. Figure 2: Are the arrows meant to line up with (phyto→PON) and (zoop→DOP) or is that just a coincidence?

This is just a coincidence

4f. Appendix Equation A34: What are $g_{max}$ and $K_{1/2}$ for the two microzoo types? Are all parameters held constant between the model version?

These values are now provided in Tables 2 and 3 of the main text. $g_{max}$ is determined by allometry but $K_{1/2}$ is not. Therefore, $g_{max}$ was different for the two microzooplankton whereas $K_{1/2}$ was not.

4g. Table A3: The parameter values pertaining to zooplankton mortality and grazing should be given here since that is the focus of the paper. Also, I could not find these parameters on the website listed. Please give the filename and folder/directory with these values.

Tables 2 and 3 provide the relevant mortality and grazing parameter values, and an exhaustive list of parameter values is provided in our online repository (link https://github.com/werdna-spatial/GUD_closure/tree/main/Paper_Data).

4h. Table A3. "The large phytoplankton has a faster maximal growth rate and higher nutrient half-saturation constants than the small phytoplankton, representative of differences in growth rate between a eukaryotic algae and a cyanobacteria, respectively (Litchman et al. 2007; Ward et al. 2012)." Is this true of the microzooplankton when there are two size classes?

The larger zooplankton actually has a slower maximal growth rate due to the allometric scaling reported in Table 3

5. Other

5a. L102-104: How do the model results presented in this paper compare to observations in the Pacific (e.g. Follett et al. 2022)? This seems relevant as much of the early literature cited on linear vs. quadratic mortality were trying to capture the difference in the seasonal cycle between the N Pacific and N Atlantic.

We have searched for additional data but have struggled to find a dataset that is suitable. Much of the data used here are from the Atlantic Meridional transects and the English Channel L4 time series. In both cases, total phytoplankton and zooplankton were taxonomically identified and counted. The Hawaii ocean timeseries filters by size but microzooplankton and phytoplankton are within the same size class, making it challenging to separate them. The Follett dataset had abundances of *Prochlorococcus*, *Synechococcus*, and heterotophic bacteria, but we were unable to locate coincident measurements of other phytoplankton and microzooplankton biomass densities. There are some existing compilations of microzooplankton biomass data (e.g. https://doi.pangaea.de/10.1594/PANGAEA.779970) but it is unclear if these have coincident phytoplankton data. Even if such data existed, biases can be introduced converting from individuals to carbon biomass density and we could not be sure that consistent methodology was used across datasets. We have therefore refrained from evaluating model performance in the Pacific but recognize that this is an important priority for future analyses.

5b. Figure 3 and results: Why just surface biomass? Aren't plankton distributed throughout the euphotic layer? Subsurface biomass could show different patterns. I suggest analyzing and depicting depth-integrated biomass.

All our biomass and productivity maps are now depth integrated

5c. Figure 3 and results: Could you please also show the microzooplankton biomass across the different models? I am curious how much it showed similar patterns to the phytoplankton, which would be expected for the quadratic closure, but not for the linear. This would help understanding the spatial differences that can't be seen in Fig 5.

These results are now shown in Figure S2 and this intuition is exactly correct – the linear closure introduces qualitative differences in spatial distribution of phytoplankton and zooplankton that go away with quadratic closure

5d. L230: Linear mortality resulting in oscillations is known behavior. Please mention in the discussion and refer to the relevant literature (e.g. Steele & Henderson 1992, Fasham 1995, Edwards & Brindley 1999, Edwards & Yool 2000).

We felt it was appropriate to recognize these prior works in the results section, when the behavior is first mentioned (Lines 254-257)

*"Consistent with prior analyses (Steele and Henderson 1992; Fasham 1995; Edwards and Brindley 1999; Edwards and Yool 2000) models with linear zooplankton losses predicts oscillations in phytoplankton and microzooplankton biomass."*

5e. Figure 6 and results: The statement that "Linear losses on the microzooplankton predict cyclic behavior in the predator-prey relationship that are inconsistent with observations" is not accurate. The slope of (a) and (b) are inconsistent with the global pattern, which are snapshots in time, but it is possible that a similar seasonal pattern exists in the observations. You would need to show those. Given the wealth of data in the English Channel from the CPR, I assume this is possible.

The English Channel data are strikingly linear and have a very high correlation coefficient. We now show these data alongside the model results in Figure 6. None of the models predict regression slopes that exactly match with the regression slope at L4, but of the simulations shown, both models with the quadratic closure are closer to the environmental data. More importantly, the level of correlation between predators and prey is much higher in models with the quadratic closure (r =0.824 and 0.834 for the parallel and diamond models, respectively) than the linear closure (r =0.25 and 0.18 for the parallel and diamond models, respectively), and the quadratic closure is much closer to the level of correlation in the L4 data (r=0.693). Predator-prey cycles tend to drive the correlation between predators and prey low, and these cycles are clearly more pronounced in the model with linear closure than the model with quadratic closure. These results are discussed in Lines 251-267.

Technical corrections:

Appendix Equations A1-A10: These are all the same form and don't all need to be shown. You could simply use an "X" or something to denote the nutrient on DOM. Also, A1 has NO3 in the equation, but should be NH4 throughout.

Appendix Equations A11-A14: Again, could just use one example of POM.

We have made these changes – thank you for the helpful suggestions

References cited:
Aumont, O., Ethé, C., Tagliabue, A., Bopp, L., & Gehlen, M. (2015). PISCES-v2: An ocean biogeochemical model for carbon and ecosystem studies. Geoscientific Model Development, 8, 2465–2513. https://doi.org/10.5194/gmd-8-2465-2015

Aumont, O., Maury, O., Lefort, S., & Bopp, L. (2018). Evaluating the potential impacts of the diurnal vertical migration by marine organisms on marine biogeochemistry. Global Biogeochemical Cycles, 32(11), 1622–1643. https://doi.org/10.1029/2018GB005886

Edwards, A. M., & Yool, A. (2000). The role of higher predation in plankton population models. Journal of Plankton Research, 22(6), 1085-1112.

Fasham,M.J.R. (1995) Variations in the seasonal cycle of biological production in subarctic oceans: A model sensitivity analysis. Deep-Sea Res. I, 42, 1111–1149.

Kearney, K. A., Bograd, S. J., Drenkard, E., Gomez, F. A., Haltuch, M., Hermann, A. J., et al. (2021). Using global-scale Earth system models for regional fisheries applications. Frontiers in Marine Science, 1121. https://doi.org/10.3389/fmars.2021.622206

Luo, J. Y., Stock, C. A., Henschke, N., Dunne, J. P., & O'Brien, T. D. (2022). Global ecological and biogeochemical impacts of pelagic tunicates. Progress in Oceanography, 205, 102822. https://doi.org/10.1016/j.pocean.2022.102822

Petrik, C. M., Luo, J. Y., Heneghan, R. F., Everett, J. D., Harrison, C. S., & Richardson, A. J. (2022). Assessment and constraint of mesozooplankton in CMIP6 Earth system models. Global Biogeochemical Cycles, 36, e2022GB007367. https://doi.org/10.1029/2022GB007367

Sieburth, J. M., Smetacek, V., & Lenz, J. (1978). Pelagic ecosystem structure: Heterotrophic compartments of the plankton and their relationship to plankton size fractions 1. Limnology and oceanography, 23(6), 1256-1263.

Stock, C., & Dunne, J. (2010). Controls on the ratio of mesozooplankton production to primary production in marine ecosystems. Deep Sea Research Part I: Oceanographic Research Papers, 57(1), 95-112.

Stock, C. A., Dunne, J. P., & John, J. G. (2014). Global-scale carbon and energy flows through the marine planktonic food web: An analysis with a coupled physical-biological model. Progress in Oceanography, 120, 1–28. https://doi.org/10.1016/j.pocean.2013.07.001

Yool, A., Popova, E. E., & Anderson, T. R. (2011). Medusa-1.0: A new intermediate complexity plankton ecosystem model for the global domain. Geoscientific Model Development, 4(2), 381– 417. https://doi.org/10.5194/gmd-4-381-2011

Yool, A., Popova, E. E., & Anderson, T. R. (2013). MEDUSA-2.0: An intermediate complexity biogeochemical model of the marine carbon cycle for climate change and ocean acidification studies. Geoscientific Model Development, 6(5), 1767–1811. https://doi.org/10.5194/ gmd-6- 1767-2013

**Reviewer 2**

Talmy et al. "Killing the predator: impacts of top-predator mortality on global-ocean ecosystem structure"

This is a nice, concise paper analyzing the effects of two variants of a plankton food web structure, with two types of losses (linear and quadratic) for the top predator in the modeled food web. Here, the predators are microzooplankton, but I believe the results are extensible to a case where there is one microzooplankton and one mesozooplankton. In Talmy et al., the authors show that the "diamond" food web structure, where one zooplankton feeds on two phytoplankton, results in a marine ecosystem with less dynamic range between the gyres and the poles in phytoplankton carbon, as well as less co-existence in community composition. Simultaneously, the quadratic losses in the top predators results in a phytoplankton to zooplankton biomass relationship that better represents recent observations. I believe This paper is written in a clear and consise way, but one major criticism is its lack of robust

engagement in other mechanisms that may contribute to variations in community co-existence, Z:P ratio, and phytoplankton carbon. Further, while the authors do cite other publications (Ward et al. 2012, Dutkiewicz et al. 2020) that list parameter values, it was very difficult to evaluate the performance of the model without a list of parameter values.

Thank you for your constructive review. We now provide a subset of parameter values in the main text, as well as a fully exhaustive list in the online code repository. We have also conducted additional simulations exploring the sensitivity of our findings to different mechanisms that contribute to community coexistence and variation in Z:P and planktonic carbon. Of particular interest to us was different assumptions about microzooplankton feeding, which we discuss in detail below and in our revised text.

Therefore, my recommendations are:

- List out in the appendix all the ecosystem-relevant parameter values used in the model

  A fully exhaustive list of parameter values is now provided in our online code repository. We have added the most relevant ecosystem parameters to the main text in Tables 2 and 3, and in the Appendix.

- Can the authors engage a bit more robustly in the discussion, other strategies that modelers use to modify zooplankton grazing in a diamond food web structure that may also result in improvements in the two main metrics (dynamic range between gyres and poles in phytoplankton carbon, community co-existence):

  We recognize that the suggestion was only to enhance the discussion and not to conduct further analysis / simulation. However, in reading Rohr et al., 2022 and other literature on this topic, we felt it was necessary to consider whether the effect of different grazing formulations (in particular Holling III) might modify our main conclusions about mechanisms contributing to biomass scaling.

  1. For example, many biogeochemical models utilize either different maximum grazing rates for zooplankton depending on the prey type, OR use a prey selectivity factor to modulate grazing. For the former, BEC/MARBL (Moore et al. 2004, Long et al. 2021) uses different maximum grazing rates for a single "adaptive" zooplankton to mimic the effect of multiple zooplankton in a single zooplankton class. For the latter, there are multiple examples of this strategy within the biogeochemical models, e.g., PISCES (Aumont et al. 2015), and COBALT (Stock et al. 2014), though those two models also have multiple zooplankton types so it may be slightly harder to compare with a single zooplankton type. However, mathematically, the effect of both these strategies would be similar.

  We appreciate drawing our attention to these common structures in ESMs. We have added some text to the discussion identifying these mismatches between our models and the ESMs, Lines 318-326:

  *"The food-web model structures assumed here are so simple that they exclude many mechanisms already considered in extant Earth system models. For example, many Earth system models contain representations of both microzooplankton and mesozooplankton. In some cases, these are explicitly represented with multiple state-variables (Stock et al. 2014; Aumont et al. 2015). In others, a single 'adaptive' zooplankton class mimics the effects of micro- and meso-zooplankton by feeding differently on phytoplankton prey types (Moore et al. 2004; Long et al. 2021). Future*

*studies may evaluate the impact of different closures in the context of these more sophisticated structures. Despite the simplicity of our models, we anticipate that our central conclusions will hold in a more general setting. Specifically, assumptions made about highest predator mortality constrain biomass scaling relationships regardless of model predictions about community composition within a trophic level."*

Here, we have agreed with your assertion (above) that there is good reason to believe our results will extend to these other ecosystem structures. There is prior literature on both the diamond (e.g. Klausmeier and Litchman 2012) and parallel (e.g. Follett et al. 2022) models indicating that, when one or two predators have mismatching grazing rates on different prey, this modifies the abundance of the prey when present, but does not modify the core predictions concerning coexistence and competitive exclusion. We have left the discussion of this topic relatively light to maintain the accessibility of our initial submission.

2. On grazing, values for the zooplankton maximum grazing rates and the grazing half-saturation constant are amongst the least well constrained parameters in food web models, and variations in these parameters have an enormous impact. Rohr et al. 2022 (Progress in Oceanography) shows this quite nicely in a robust analysis, along with evaluating differences in the grazing functional form itself (Holling type II or type III functional responses). It would be nice if the authors could engage a bit more in the discussion regarding whether modelers would be able to compensate for the lack of a second zooplankton (e.g., in the diamond food web model) by modulating maximum grazing rates and grazing half-saturation constants.

We feel that the issue of Holling II vs. III, and the ability of the diamond food-web to compensate for the lack of a second zooplankton with respect to phytoplankton coexistence, are to some extent different questions. In the context of this paper, we have already shown that the impact of the closure holds true regardless of model predictions about phytoplankton community composition. Therefore, even if the adaptive feeding assumed in many ESMs could mimic the effect of a second zooplankton, we would not anticipate that it would impact the biomass scaling relationships predicted by the model. On the other hand, based on Rohr et al., 2022 and other literature, it does seem likely for Holling III to impact our predictions regarding Z vs. P biomass scaling. We expand on both of these issues here but have kept our discussion of these topics in the main text relatively light, to preserve the conciseness that both reviewers commented on.

Regarding the coexistence issue, as we note above, allowing the microzooplankton to be 'adaptive' in the manner assumed in many ESMs by modifying max grazing rates and half-saturation values is unlikely to fully account for a second zooplankton with respect to phytoplankton coexistence, as these grazing parameters modify phytoplankton abundances but do not allow the same degree of coexistence. Interestingly, allowing zooplankton to actively switch prey type based on prey abundance (following Vallina et al., 2014) does provide some degree of coexistence in the phytoplankton but not to the same degree as parallel feeding (compare columns in Figure S7). We report these results in Lines 284-286:

*"allowing microzooplankton to switch actively to prey on more abundant phytoplankton allows greater coexistence of phytoplankton in the diamond food web (Figure S7 and Vallina et al. 2014) but unsurprisingly does not qualitatively modify the scaling relationships reported in Figure 5."*

We were very interested in the Rohr et al., 2022 study as it pertains to our main predictions about mechanisms driving Z vs. P biomass scaling. Interestingly, when we assume Holling III instead of Holling II, we find significantly more correlation between Z and P for all closures (compare Figure S8 of the SI to Figure 5 of the main text). Even when the linear closure is assumed, Holling III predicts correlation between Z and P that is much closer to the environmental data (Figure S8a,b). Nevertheless, when Holling III is assumed, the quadratic closure still predicts more realistic levels of correlation between Z and P. These new results are explained in Lines 286-292:

*"Microzooplankton feeding according to type III functional response leads to far greater correlation between predators and prey across models (Figure S8). These results are consistent with prior studies identifying type III feeding as a stabilizing mechanism on microzooplankton-phytoplankton dynamics (Rohr et al., 2022). Nevertheless, even when a type III response is assumed, the quadratic closure still leads to more realistic correlation between microzooplankton and phytoplankton than the linear closure (Figure S8) pointing to the closure as an important control on trophic structure globally."*

Interestingly, the regression slopes with Holling III are all unrealistically steep (figure S8), which we believe is because variance in P is significantly reduced due to the Holling III introducing a refuge in grazing rate at low prey abundance, preventing phytoplankton abundance from reaching very low values. At this junction, we cannot say if the unrealistically steep regression slope introduced by Holling III is an artefact, as it's possible that mesozooplankton predation on microzooplankton also follows a Holling III response, which we anticipate would bring the model slopes back in line with the observational data. This leads to several novel research questions: 1) does mesozooplankton grazing on microzooplankton follow Holling III? 2) if so, does introducing mesozooplankton feeding according to Holling III lead the diamond food-web model to predict realistic biomass scaling when Holling III is also assumed in the microzooplankton-phytoplankton interaction? and 3) can the effects of the mesozooplankton be parameterized with an appropriate microzooplankton closure? We feel these questions are all very interesting but go well beyond the scope of the current study. For parsimony we have limited our revisions to explaining the impact of Holling III on the level of correlation between Z and P (Lines 289-292 and Figure S8) which we anticipate will be a general result that holds true regardless of whether or not mesozooplankton are explicitly resolved.

3. Lastly – prey switching is a major issue that is only mentioned in passing in the discussion. It would be nice to see a more robust discussion – do the authors think that modifications in the switching form would result in substantial changes in the modeled ecosystem, and why? There are a lot of approaches towards switching, as laid out extensively in Gentleman et al.

(2003), but models typically use just one or two forms (e.g., Stock et al. 2008 Journal of Marine Systems has addressed this quite nicely in a simple system). In my opinion, a more than cursory treatment of this topic would be important in this paper.

As per our comments above, active switching promotes phytoplankton coexistence (Vallina et al., 2014 and figure S7) but does not modify biomass scaling predictions (explained in lines 284-286). We thank this reviewer for helping to direct our attention to these important questions, as we feel addressing these issues has allowed us to deepen and expand our understanding of the models addressed in this study.

4. Other parameters that may additionally modulate phytoplankton carbon in food web models that aren't addressed include the fraction of phytoplankton and zooplankton losses that go to dissolved organic matter vs. particulate organic matter, which may influence the recycling rate and strength of the microbial loop. Lastly, variations in the relative nutrient uptake rate of the different phytoplankton may also result in more or less differences in the phytoplankton carbon between the gyres and the poles.

We now include sensitivities to phytoplankton size, which modify nutrient affinities (through allometry) but do not modify biomass scaling relationships (lines 282-283). Also, we have conducted sensitives to the POM-DOM partitioning (referred to in our model as an 'export fraction'). Results show that, when modified over a reasonable range, the average predator-prey scaling is sensitive to this number, but the scaling relationship is not. We have opted not to include these results as they go beyond the scope of the present study, which has expanded significantly in response to these helpful reviews.

Other than these points, I found the manuscript written quite clearly, with compelling figures and nice presentation. With a more robust discussion addressing a range of these additional points listed above, this manuscript would make a nice addition to the literature.

We thank this reviewer for the careful reading of our manuscript, and their helpful suggestions. We feel our manuscript has improved significantly as a result.

---

## Referee Report (RR1)

**Reviewer 1**

General comments:

I thank the authors for their careful consideration of reviewers' comments and their thorough revision. All of my specific comments have been satisfactorily addressed, except for comment "4b" about plankton sizes. The microzooplankton predators are smaller than their large phytoplankton prey, which is unrealistic in size-structured marine food webs. Otherwise, the paper addresses relevant scientific questions within the scope of BG. Though the concepts, ideas, and tools are not necessarily novel, their implementation and the comparison with recent observational datasets are. Overall, I think is an elegant study that would be valuable to the BG readership.

(Original) Specific comments:

4. Missing information on experiment design and parameters.

4b. L114: What is the sensitivity of the results to these assumed sizes? I would argue that when ESMs use only one zooplankton type, it is supposed to represent both micro and mesozoo. Similarly, when there are two zoo types, one is micro and one is meso, and the meso preys on the micro, which is missing from the parallel food web here. Also, the "large phytoplankton" here is barely the size of the diatoms in ESMs (10-100 um, "microplankton"), while both the small and large microzooplankton are also at the low end of the microzooplankton (10-200 um in ESMs, 2-200 um in Sieburth et al. 1978), and when there is only one zooplankton group in ESMs it tends to encompass everything 10-2,000 um. I do not mean to suggest that this study is without value for that reason, but the comparison to global ESMs used for climate change studies is less direct.

Interestingly, there seems to be a great deal of variation regarding what is being described in ESMs. From Rohr et al., 2023: "*it is concerning that some models imply something statistically similar to an ocean filled entirely with very slow-grazing meroplankton larvae (MEDUSA2.1, OECOv2) and others an ocean filled entirely with very rapidly-grazing ciliates (MARBL and CMOC).*" Therefore, our modeled rates of grazing are likely to fall within the envelope of values simulated in ESMs. Nevertheless, we conducted sensitivity simulations modifying phytoplankton and zooplankton size (described in lines 174-179 and lines 282-283) and found that our primary conclusions regarding scaling relationships are insensitive to assumptions about size. These results are shown in Figure S6.

I appreciate the size sensitivity test. However, I still find the microzooplankton sizes to be unreasonable. How could a 3.2 um or 4.7 um ESR ciliate feed on a 7 um ESR large phytoplankton? Could you change your results presented in the main paper to those with the larger microzooplankton from the sensitivity test?

4e. Figure 2: Are the arrows meant to line up with (phyto→PON) and (zoop→DOP) or is that just a coincidence?

This is just a coincidence.

Could you move the arrows to prevent confusion?

Other:

Table 3: I assume these size-dependent parameters take the form: parameter = a * size ^ b. But what is size? The ESR given in Section 2.1? Please specify.

Line 269-273. Is the Z:P ratio in terms of biomass? The *z-ratio* (as in Stock et al. 2014) is productivity. Please specify.

Technical corrections:

Figure S5 is missing the units.

---

## Author Response (AR2)

**Reviewer 1**

General comments:

Talmy et al. Biogeosci

I thank the authors for their careful consideration of reviewers' comments and their thorough revision. All of my specific comments have been satisfactorily addressed, except for comment "4b" about plankton sizes. The microzooplankton predators are smaller than their large phytoplankton prey, which is unrealistic in size-structured marine food webs. Otherwise, the paper addresses relevant scientific questions within the scope of BG. Though the concepts, ideas, and tools are not necessarily novel, their implementation and the comparison with recent observational datasets are. Overall, I think is an elegant study that would be valuable to the BG readership.

**We have changed our plankton sizes to make them more ecologically realistic. Please see specific comments below for more detail. We greatly appreciate this reviewer's attention to detail.**

(Original) Specific comments:
4. Missing information on experiment design and parameters.

4b. L114: What is the sensitivity of the results to these assumed sizes? I would argue that when ESMs use only one zooplankton type, it is supposed to represent both micro and mesozoo. Similarly, when there are two zoo types, one is micro and one is meso, and the meso preys on the micro, which is missing from the parallel food web here. Also, the "large phytoplankton" here is barely the size of the diatoms in ESMs (10-100 um, "microplankton"), while both the small and large microzooplankton are also at the low end of the microzooplankton (10-200 um in ESMs, 2- 200 um in Sieburth et al. 1978), and when there is only one zooplankton group in ESMs it tends to encompass everything 10-2,000 um. I do not mean to suggest that this study is without value for that reason, but the comparison to global ESMs used for climate change studies is less direct.

Interestingly, there seems to be a great deal of variation regarding what is being described in ESMs. From Rohr et al., 2023: "*it is concerning that some models imply something statistically similar to an ocean filled entirely with very slow-grazing meroplankton larvae (MEDUSA2.1, OECOv2) and others an ocean filled entirely with very rapidly-grazing ciliates (MARBL and CMOC).*" Therefore, our modeled rates of grazing are likely to fall within the envelope of values simulated in ESMs. Nevertheless, we conducted sensitivity simulations modifying phytoplankton and zooplankton size (described in lines 174-179 and lines 282-283) and found that our primary conclusions regarding scaling relationships are insensitive to assumptions about size. These results are shown in Figure S6.

I appreciate the size sensitivity test. However, I still find the microzooplankton sizes to be unreasonable. How could a 3.2 um or 4.7 um ESR ciliate feed on a 7 um ESR large phytoplankton? Could you change your results presented in the main paper to those with the larger microzooplankton from the sensitivity test?

**We have revised our plankton sizes as follows (see Lines 116-121):**

*"In both formulations, small and large phytoplankton represent cells with ~0.5 and 5µm equivalent spherical radius, and are representative of picocyanobacteria and eukaryotic algae, respectively. In the parallel model, small and large microzooplankton represent protists with ~7 and 50µm equivalent spherical radius and are representative of microzooplankton in the ciliate size range. The generalist predator in the diamond food-web model has 15µm equivalent cell radius."*

**Using the allometric scaling for maximal grazing rates in Table 3, the large microzooplankton had unrealistically low grazing rates with these new sizes. We revisited our allometric coefficients and realized that we had compensated for unrealistically small microzooplankton with unrealistically small values for 'a' (where grazing rate is a function of cell volume with aV$^b$). Similarly, our grazing half saturation rate (which is size independent) was too high (see Table 2). We have modified these values to more closely reflect the published literature (Hansen et al. 1997; Ward et al. 2012; Rohr et al. 2022). The resulting grazing rates are very close to those use in our previous submission, but we have rerun all simulations and we do see some modest changes, for example with slightly greater contributions of the large phytoplankton in the parallel food web (deeper blue colors in Figure 4). However, none of our findings are modified by the changes.**

4e. Figure 2: Are the arrows meant to line up with (phyto→PON) and (zoop→DOP) or is that just a coincidence?

This is just a coincidence.

Could you move the arrows to prevent confusion?

**The arrows now both point to the top of the detritus box**

Other:

Table 3: I assume these size-dependent parameters take the form: parameter = a * size ^ b. But what is size? The ESR given in Section 2.1? Please specify.

**Table 3 caption now says 'Coefficients a and b constrain allometric relations of the form aV$^b$ where V represents cell volume (µm$^3$)'**

Line 269-273. Is the Z:P ratio in terms of biomass? The *z-ratio* (as in Stock et al. 2014) is productivity. Please specify.

**This paragraph (now lines 284-294) and Figure 7 caption now refer to the Z:P biomass ratio throughout**

Technical corrections:
Figure S5 is missing the units.

**The figure now has units**

Hansen, P. J., P. K. Bjornsen, and B. W. Hansen. 1997. Zooplankton grazing and growth: Scaling within the 2-2,000-micrometer body size range. Limnol. Oceanogr. 42: 687–704.

Rohr, T., A. J. Richardson, A. Lenton, and E. Shadwick. 2022. Recommendations for the formulation of grazing in marine biogeochemical and ecosystem models. Prog. Oceanogr. 208: 102878.

Ward, B. A., S. Dutkiewicz, O. Jahn, and M. J. Follows. 2012. A size-structured food-web model for the global ocean. Limnol. Oceanogr. 57: 1877–1891.